# A retrofit sensing strategy for soft fluidic robots

Shibo Zou[1], Sergio Picella[1,2], Jelle de Vries[1], Vera G. Kortman [3,4], Aimée Sakes[4] & Johannes T. B. Overvelde [1,2] ✉

Soft robots are intrinsically capable of adapting to different environments by changing their shape in response to interaction forces. However, sensory feedback is still required for higher level decisions. Most sensing technologies integrate separate sensing elements in soft actuators, which presents a considerable challenge for both the fabrication and robustness of soft robots. Here we present a versatile sensing strategy that can be retrofitted to existing soft fluidic devices without the need for design changes. We achieve this by measuring the fluidic input that is required to activate a soft actuator during interaction with the environment, and relating this input to its deformed state. We demonstrate the versatility of our strategy by tactile sensing of the size, shape, surface roughness and stiffness of objects. We furthermore retrofit sensing to a range of existing pneumatic soft actuators and grippers. Finally, we show the robustness of our fluidic sensing strategy in closed-loop control of a soft gripper for sorting, fruit picking and ripeness detection. We conclude that as long as the interaction of the actuator with the environment results in a shape change of the interval volume, soft fluidic actuators require no embedded sensors and design modifications to implement useful sensing.

The intrinsic compliance of soft robots provides adaptability to unknown environments[1–3]. For example, a soft robotic gripper passively adapts its body shape, making it possible to grasp various objects without the need for active sensing[4,5]. However, when it comes to more advanced tasks such as identifying and sorting objects, sensory feedback from the gripper becomes essential to achieve closed-loop control in gripping and manipulation[6]. Benefiting from advances in soft materials, soft robotic sensing has been enabled by embedding flexible or stretchable sensors made from piezoresistive and piezocapacitive polymer composites[7], liquid metals[8], electrically and ionically conductive hydrogels[9], and polymeric optical waveguides[10]. Both proprioception (sensing of self-deformation) and exteroception (sensing of external stimuli) of soft robots have been successfully demonstrated with embedded sensors. Moreover, multimodal sensing, i.e., the simultaneous perception of multiple physical parameters,

has been achieved by machine learning[11,12] and embedding various sensors into the soft actuator[13–15]. A common feature in all these sensing strategies for soft robotic applications is the separation of actuation and sensing elements[16,17]. This is a result of the compliance of the soft systems, which complicates integration and reduces reliability of the sensors that need to be embedded in the soft actuator, therefore placing considerable constraints on the design of both the sensors and actuators.

As fluidic actuation represents a plurality in soft robotics[18], sensing strategies based on fluidic media, either gas or liquid, have been investigated to reduce the integration difficulties of actuation and sensing elements, such as fluidic resistance sensing[19,20], fluidic pressure sensing[21–32] and electrical resistance sensing[33]. Most fluidic sensing strategies incorporate an additional cavity in the soft actuator[21,24–26,28,29]. Since the enclosed cavity contains a fixed amount of

---

[1]Autonomous Matter Department, AMOLF, Amsterdam 1098 XG, The Netherlands. [2]Institute for Complex Molecular Systems and Department of Mechanical Engineering, Eindhoven University of Technology, Eindhoven 5600 MB, The Netherlands. [3]Department of Marine and Transport Technology, Delft University of Technology, Delft 2628 CD, The Netherlands. [4]Bio-Inspired Technology Group, Department of BioMechanical Engineering, Delft University of Technology, Delft 2628 CD, The Netherlands. ✉e-mail: overvelde@amolf.nl

fluid, deformation of the actuator or contact with the environment changes the volume of the cavity and thus increases or decreases the internal pressure. Interestingly, this pressure response can be measured remotely by connecting the cavity and electronic pressure sensor via a tube, such that no electronic components need to be embedded in the soft actuator.

A particularly interesting yet simple method uses the cavity of the soft actuator itself to sense external force by measuring and analyzing the fluidic pressure of the soft actuator[34,35]. The benefit of such a self-sensing approach has also been demonstrated in dielectric elastomer actuators[36–38] and electrohydraulic actuators[39–41]. In these systems the electrical characteristics of the actuator can be measured to infer the mechanical deformation while it is being actuated, hence no additional sensors and associated electronics are needed[37]. While fluidic self-sensing has originally been demonstrated in a potential medical application[34], a natural question to ask is how widely applicable, versatile and robust such an approach is. To answer this question, we need to gain a better understanding of the underlying principles that allow for fluidic self-sensing, and determine if we can infer the interaction of a wide variety of soft actuators with their environment by measuring and analyzing the fluidic response of the enclosed cavity. And if so, we want to determine how easy it is to integrate and retrofit such a sensing approach, and if interactions with the environment can be robustly measured.

To achieve this, in this work we will first experimentally show how the response of a typical soft fluidic bending actuator changes when interacting with the environment. We next introduce several strategies to sense these interactions without the need to embed additional sensing elements in the soft actuator. We demonstrate how to apply our fluidic sensing strategy to a soft gripper, and to enable a versatile range of sensing applications such as size, shape, surface roughness

and stiffness sensing of objects. To demonstrate that the sensing approach can be retrofitted, we apply the sensing strategy to a filament actuator, a McKibben actuator, a thermoplastic polyurethane (TPU) actuator, a soft suction gripper specifically designed for medical applications and two commercially available soft grippers. We furthermore developed a basic model based on a linear extension actuator to study the underlying factors that determine the sensing resolution. Finally, we show that our fluidic sensing strategy is robust enough to implement closed-loop control in gripping and sorting applications.

## Results

### Fluidic sensing of the soft robot-environment interaction

We start by looking into the characteristic behavior of a typical soft PneuNet bending actuator[42] when interacting with the environment. We inflate a soft actuator onto a rigid plate from different heights $h$, and characterize the pressure-volume response for each height, i.e., the pressure $P$ as a function of supplied air volume $V$, at different heights (fitting curves in Fig. 1a, test results in Fig. S1, test procedure in Methods section Distance Sensing with PneuNet Bending Actuator and Supplementary Video 1). Interestingly, the interaction with the plate influences the pressure-volume response of the soft actuator. This influence originates from the compliance of the soft actuator and the effect that external forces have on the internal geometric volume of the inflated actuator. According to the ideal gas law, the difference in internal geometric volume gives a direct fluidic response in the form of a variation in pressure, if the temperature is constant and the amount of air at different heights is equal. Importantly, by definition any physical interaction with the environment leads to a change in the internal geometric volume of a soft actuator because of the compliance of the soft body.

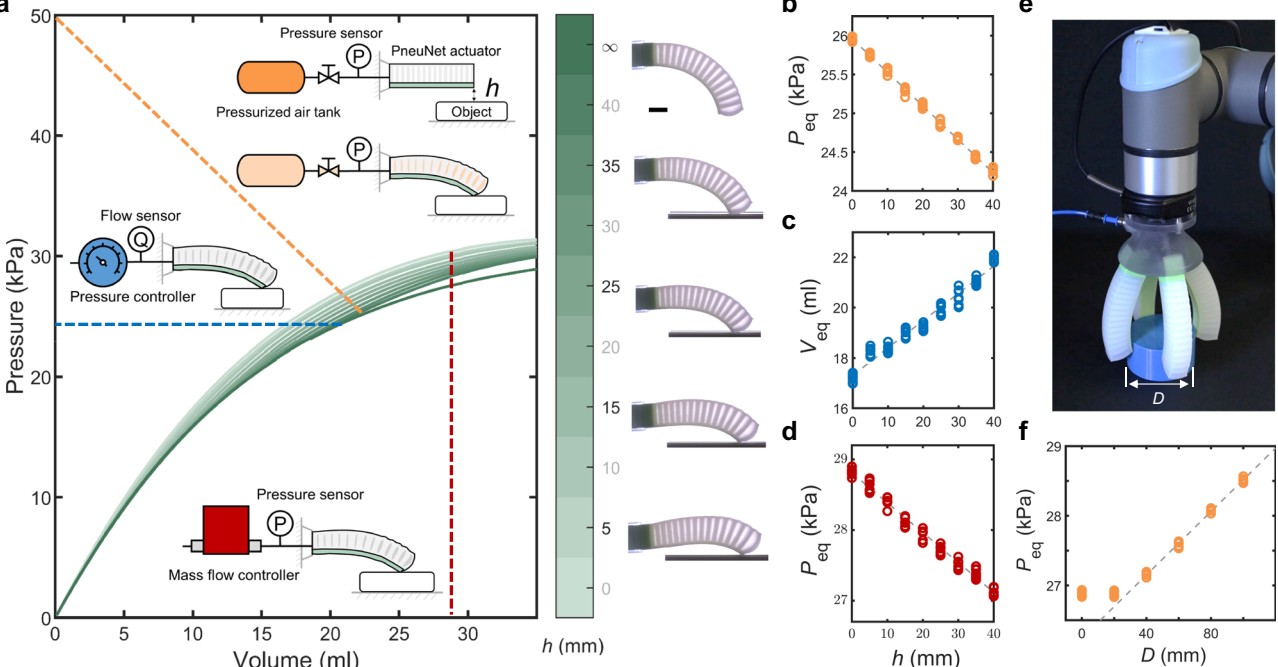

**Fig. 1 | Fluidic sensing of the soft robot-environment interaction. a** Overview of the fluidic sensing methodology. Inflating a PneuNet actuator onto a rigid plate from a height $h$ changes the pressure-volume response of the soft actuator, which can be characterized by three different fluidic sensing methods: pressurized air tank and solenoid valve (orange dashed line), pressure control (blue dashed line), and flow control (red dashed line). For clarity, the presented data is fitted based on measured results shown in Fig. S1. The snapshots represent the equilibrium state of the soft bending actuator with an input volume of 28.7 ml at $h = 5, 15, 25, 35$ and $\infty$

mm. Scale bar: 20 mm. **b-d** Calibrations between the height $h$ and the equilibrium pressure $P_{eq}$ or total input volume at equilibrium $V_{eq}$ based on sensing with the three fluidic sensing methods (colors match sensing method). The dashed lines represent the linear fits of all the test data. **e** A soft gripper with four PneuNet actuators gripping a cylindrical object with a diameter $D$. **f** Calibration between the cylinder diameter $D$ and the equilibrium pressure $P_{eq}$ of the soft gripper. The dashed line represents a linear fit of the test data with $D = 40, 60, 80, 100$ mm.

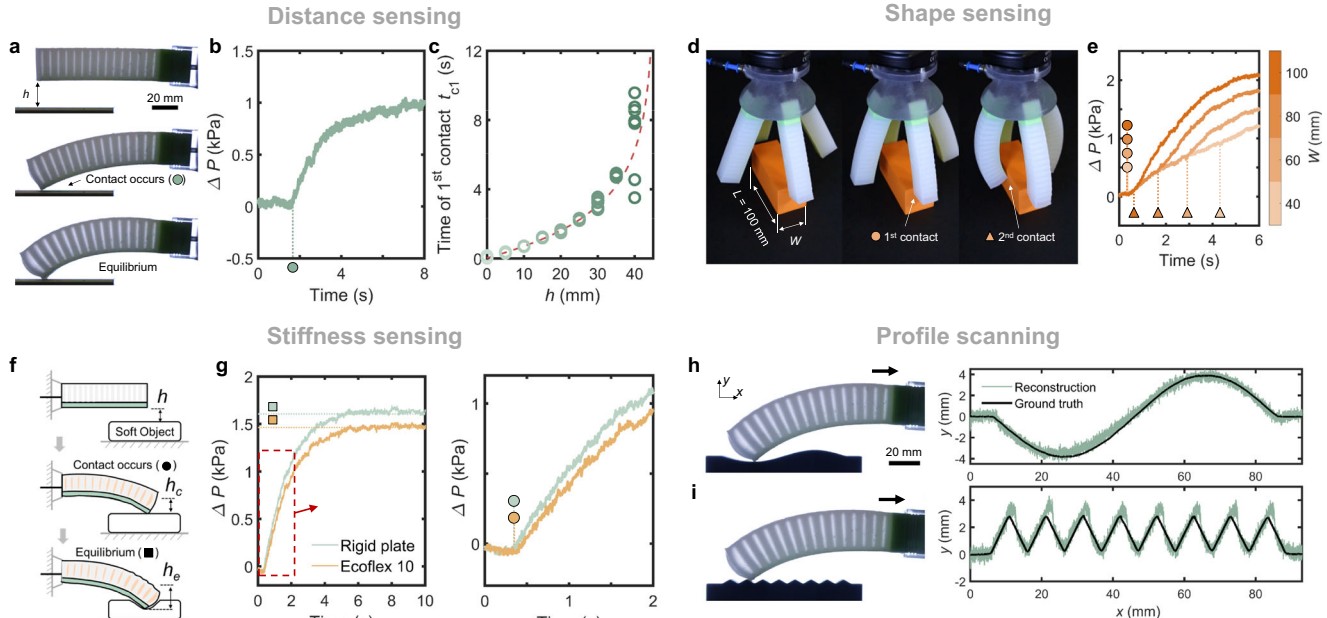

**Fig. 2 | Time-enabled sensing versatility.** The difference in fluidic pressure response $\Delta P$ between the soft actuator with and without environmental interaction is evaluated over time to implement sensing. **a-c** Inferring the initial distance $h$ between the actuator and plate from the time of first contact $t_{c1}$. $h = 20$ mm in **(a)** and **(b).** The dashed line in **c** represents the vertical displacement-time curve of the tip of actuator in the case of free actuation. **d, e** Shape sensing of rectangular objects by measuring both the time of first and second contact of the soft gripper. **f, g** Stiffness sensing by measuring both the time of first contact and the equilibrium pressure, which indicate the vertical displacement of the actuator at the first contact ($h_c = h$) and equilibrium ($h_e \geq h$ depending on the object stiffness), respectively. $h = 5$ mm in **(g). h, i** Surface scanning and profile reconstruction of two different surface profiles.

In order to effectively sense these differences in the pressure-volume response, and with that the interaction of the soft actuator with the environment, we connect the soft actuator to a pressurized air tank via a solenoid valve and measure the equilibrium pressure using an external pressure sensor after opening of the valve (orange line in Fig. 1a). Depending on the initial distance $h$ of the actuator to the surface, the equilibrium pressure $P_{eq}$ will be slightly different. Interestingly, for this specific actuator design and interaction, the relationship between the equilibrium pressure $P_{eq}$ and height $h$ can be approximated by a linear relationship. Therefore, the initial distance between the actuator and plate can be inferred from the fluidic signal based on the calibration curve (Fig. 1b and Fig. S2a) with an accuracy of $\pm 1.7$ mm (Fig. S3a). The force applied by the actuator on the plate can also be inferred from the equilibrium pressure $P_{eq}$ (Fig. S4). Note that since the steel air tank has a linear pressure-volume relationship, varying the tank size effectively changes the slope of the tank's pressure-volume curve. This changes the intersection points with the actuator's pressure-volume curves in Fig. 1a, making it possible to tune the sensing resolution (Fig. S5).

While here we connect the actuator to a steel tank and solenoid valve to implement sensing, depending on the available equipment and precision requirement, the interaction of a soft actuator with the environment can also be characterized by controlling the pressure and measuring the volume flow input (blue line in Fig. 1a, c and Fig. S2b) with a sensing accuracy of $\pm 4.7$ mm (Fig. S3b), or controlling the volume flow input and measuring pressure (red line in Fig. 1a, d and Fig. S2c) with a sensing accuracy of $\pm 3.4$ mm (Fig. S3c).

Our sensing strategy can also be directly applied to a soft gripper to sense the size of objects. We demonstrate size sensing of cylindrical objects using a soft gripper consisting of four PneuNet bending actuators (Fig. 1e, Fig. S6 and Supplementary Video 1). To enable sensing, the actuators are jointly connected to the external system that contains an air tank, a solenoid valve and a pressure sensor. Figure 1f shows that when gripping larger objects, a higher equilibrium pressure $P_{eq}$ is reached. Interestingly, the relationship between the equilibrium

pressure $P_{eq}$ and cylinder diameter $D$ can also be fitted with a linear function within the grasping range of the gripper ($d \gtrsim 20$ mm). Repeated tests on a similar gripper showed the same results, where we found that the pressure measurements vary within $\pm 0.08$ kPa over 100 cycles (Fig. S7).

## Time-enabled sensing versatility

Having demonstrated the basic principles of our sensing approach, we next show that versatile sensing applications can be achieved by measuring the pressure response of the soft actuator over time. To show how we can extract more information from the pressure-time response, we first revisit height sensing where so far we only considered the equilibrium pressure at a specific moment in time (Fig. 1). Instead, if we correlate the pressure-time response of the actuator to a reference response, i.e., free actuation without interacting with the environment, we can determine the moment of contact (Fig. 2a, b). In Fig. 2b, we evaluate $\Delta P = P - P_{ref}$ over time from the onset of actuation and determine the time of first contact $t_{c1}$ when $\Delta P > 0$. We can then use $t_{c1}$ to infer the initial distance $h$ between the actuator and plate (Fig. 2c) based on the tip displacement-time curve of the actuator in the reference response, with an accuracy of -2.9 to 3.8 mm (Fig. S3d). Note that the sensing speed of this strategy is dominated by the actuation speed, which is determined by the flow resistance between the air tank and actuator (Fig. S8). Interestingly, a higher sensing response speed can be achieved by measuring the time of contact $t_{c1}$ (Fig. 2c) compared to the equilibrium $P_{eq}$ (Fig. 1b), because the measurement of $t_{c1}$ does not require the system to reach equilibrium.

Based on this approach, we show how we can sense *(i)* the shape of objects, *(ii)* the stiffness of a soft substrate and *(iii)* the profile of a surface. As a first demonstration of the sensing versatility, we use our previously introduced gripper to sense the aspect ratio of rectangular objects. Figure 2d, e and Supplementary Video 2 show that two contact events occur in the pressure-time response when the soft gripper grips a rectangular object, indicating the length and width of the object, respectively. While this could also be achieved by individually

addressing each actuator, which would likely make it easier to extract shape information from the soft gripper, it would also require additional hardware that might not be needed or available in specific applications.

As a second demonstration of the sensing versatility, we achieve multimodal sensing of distance and stiffness when the actuator interacts with a soft plate (Fig. 2f). Here, the time of first contact $t_{c1}$ extracted from the pressure-time response (Fig. 2g, Supplementary Video 2) indicates the initial distance between the actuator and plate, while the equilibrium pressure $P_{eq}$ indicates the final vertical displacement of the actuator. By comparing these two displacements, we can extract the indentation depth of the soft actuator, which can be correlated to the stiffness of the plate when considering the stiffness of the soft actuator. Therefore, measuring $t_{c1}$ and $P_{eq}$ together makes it possible to compare the stiffness values of objects (Fig. 2g).

As a final demonstration of the sensing versatility, we use the actuator as a profilometer by considering the variations in the equilibrium pressure when moving the actuator along a surface (Fig. 2h, i and Supplementary Video 2). To show this, we move the actuator horizontally along a surface with a robotic arm and measure the pressure response continuously. The profile of the object can be reconstructed using a calibration curve (Fig. 1b) and a reference pressure response, which rules out the influence of system leakage or other variations over time. Note that in this sensing application the sharpness of the tip of the soft actuator will determine the resolution of the sensing signal.

## Retrofitting the fluidic sensing approach

In all demonstrations so far, we used one or more identical soft bending actuator. However, our sensing approach can also be retrofitted to a broad range of fluidic actuators without the need for any design changes. To demonstrate the wide applicability, using our approach we sensorize a filament actuator, a McKibben actuator, a 3D-printed bending actuator[43], a suction cup, and two commercial soft grippers (Figs. 3, 4 and Supplementary Video 3).

We start by retrofitting our sensing strategy to a filament actuator[44–46] and a McKibben actuator[47,48] to sense the angular displacement of a joint in the artificial muscle demonstration[39,45]. In both cases (Fig. 3a-f), we can correlate measured equilibrium pressure to the angular displacement of the joint. We do observe that for the filament actuator less sensing resolution is achieved compared to the McKibben actuator. This is likely due to the fact that the deformation of the filament gripper is less well-defined, especially when comparing it to the McKibben actuator where the environment has a strong influence on the internal volume. We hypothesize that this stronger influence is the results of the wires that to some extent limit the degrees of freedom.

To determine if our sensing approach can also be used for higher actuation pressures, we next retrofit our sensing strategy to a 3D-printed TPU bending actuator that requires an actuation pressure around 200 kPa[43]. In previous tests with the bending actuator, we only consider a single contact between the soft actuator and the environment. Since the TPU bending actuator forms a circular shape at higher pressures[43], we tested our sensing strategy with conformal grasping[49], where the soft actuator interacts with the cylindrical object at multiple contact points (Fig. 3g). We find that the conformal grasping of cylindrical objects with various diameters results in different pressure-volume responses of the soft actuator (Fig. 3h) and that we can also correlate the equilibrium pressure with the diameter of the grasping object, even for these higher pressure ranges (Fig. 3i).

To test our retrofitting approach with soft grippers, we first apply it to a suction cup (Fig. 4a) that was specifically designed for tissue gripping in Minimal Invasive Surgery (MIS)[50]. The requirements for the foldability, adaptability and biocompatibilty of the tissue gripper make it challenging to embed sensors in the gripper itself to obtain sensory feedback during operation. With our fluidic

sensing strategy, the pressure sensor can be connected remotely to the gripper outside of the human body and no additional design change of the gripper is needed. Once vacuum is applied to the soft gripper, the connected surface gets pulled into the gripper, reducing its internal geometric volume (Fig. 4b). The surface stiffness influences the pressure-volume response of the gripper through the amount of reduced internal geometric volume of the gripper (fitting curves in Fig. 4c, and test results in Fig. S9). The final equilibrium pressure $P_{eq}$ can be used to infer the stiffness of the surface that is attached to the gripper, where the sensing resolution can be tuned by the initial pressure $P_0$ in the air tank (Fig. 4d and Fig. S10). Furthermore, we found that the equilibrium pressure in the gripper changes almost linearly with the pulling force applied on the gripper (Fig. 4e), making it possible to predict when the gripper would detach from the surface which in our experiments occured at $\Delta P \approx 1.2$ kPa with an average detaching force of 4.53 N.

Similarly, we demonstrate that we can retrofit sensing to commercial soft grippers (Fig. 4f–k), one powered by vacuum pressure[51] and the other by positive pressure[52]. Even though the pressure-volume relation for both grippers is relatively different, in both cases we find a linear correlation between the size of cylindrical objects and the measured equilibrium pressure. Note that because the internal volumes of the actuators and grippers are different in these examples, we had to replace some of the external hardware. For example, the internal volume of the air tank, which determines the slope of the tank's pressure-volume response, is chosen based on a compromise between initial tank pressure and sensing resolution (Fig. S5).

## Characterizing the sensing resolution

Figures 3 and 4 provide a general picture of how the variations in pressure-volume curves between the soft actuators and grippers lead to different sets of equilibrium points based on the conservation of air mass. The absolute pressure change observed during robot-environment interactions ranges from 1.3 kPa to 5.9 kPa among the soft actuators and grippers we tested (Table 1). Even though the relative pressure difference (pressure difference due to interaction with the environment in comparison to maximum pressure obtained in the actuator) might be smaller for actuators that require higher inflation pressure (e.g., TPU and filament actuators), the absolute pressure change for all actuators we tested is in the same order of magnitude (Table 1). Note that the average sensing resolution can be determined by dividing the absolute pressure change by the tested range of sensing target and is therefore not affected by a lower relative pressure difference.

In order to uncover the underlying factors that determine the sensing resolutions of the soft actuators and grippers, we consider both the initial and final states of the system. At the initial state, the air tank with an internal geometric volume $v_{tank}$ is pressurized at $p_{tank} = p_0$, and the actuator with an internal geometric volume $v_{act} = v_0$ is at atmosphere pressure $p_{act} = p_{atm}$. At the final state, the air tank and the actuator reach the same pressure $p_{tank} = p_{act} = p_1$, and the internal geometric volume of the actuator becomes $v_{act} = v_1$. Assuming constant temperature, since the total amount of air mass inside the system (air tank and the actuator) stays constant, according to the ideal gas law, we have

$$p_0 v_{tank} + p_{atm} v_0 = p_1 v_{tank} + p_1 v_1. \tag{1}$$

Note that the absolute pressure here is indicated in lowercase letter to distinguish it from the relative pressure (with respect to atmospheric pressure) that is used elsewhere in the manuscript. When the total amount of air inside the system remains constant, $v_1$ only depends on the interaction of the soft actuator with the environment, i.e., the

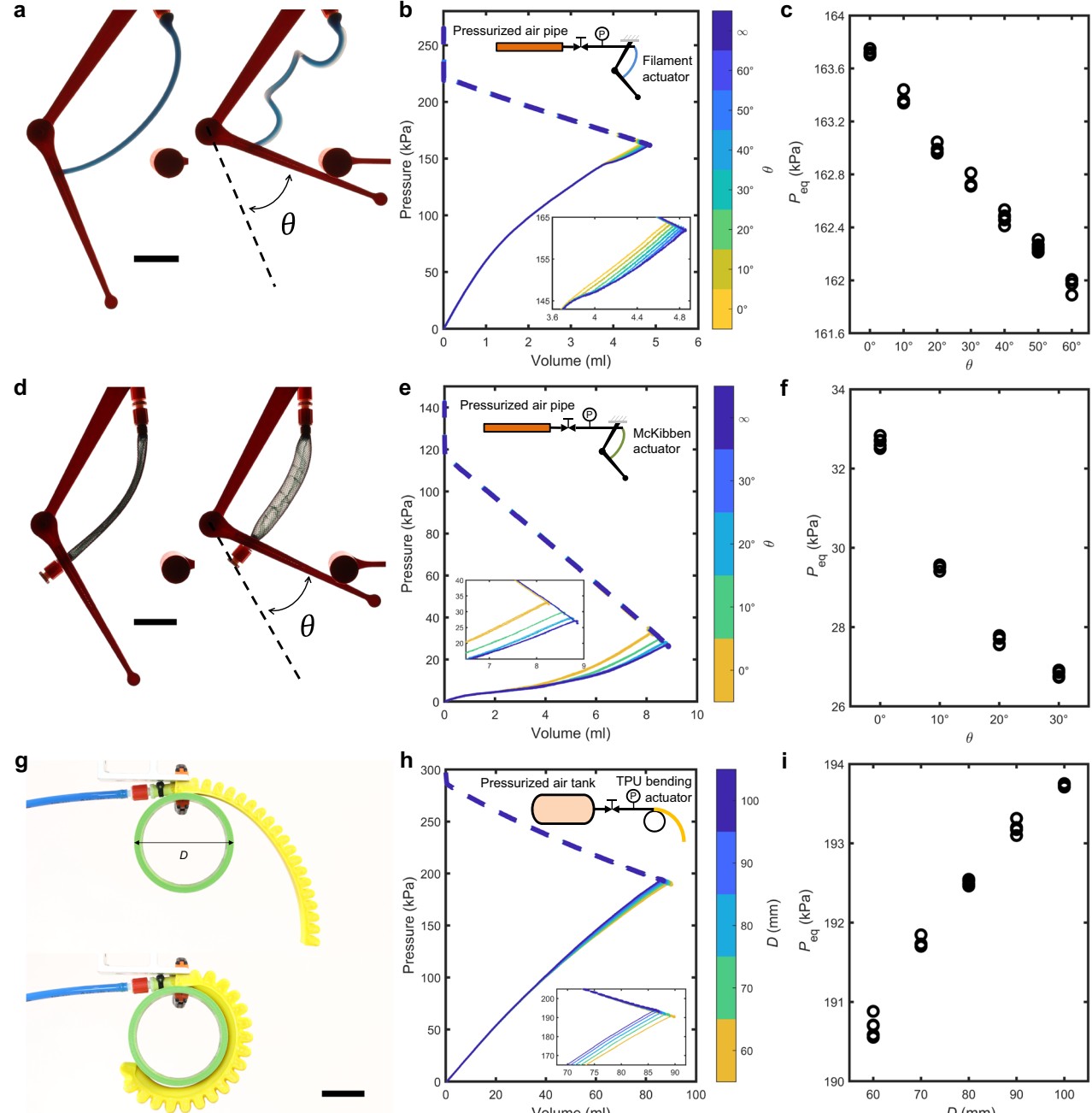

**Fig. 3 | Retrofitting the fluidic sensing approach to a filament actuator (a-c), a McKibben actuator (d-f) and a 3D-printed bending actuator (g-i).** The filament (**a**) and McKibben (**d**) actuator are used as a muscle to rotate an arm towards a stopper. TPU bending actuator (**g**) wrapping around a cylinder with a diameter *D*. Corresponding pressure-volume relation for the soft actuator (solid) and tank (dashed) (**b**, **e**, **h**) and equilibrium pressure $P_{eq}$ in the system (**c**, **f**, **i**) for different positions of the stopper or cylinder diameter *D*. Experimental results from five tests are shown for each $\theta$ (**b**, **c**, **e**, **f**) and each *D* (**h**, **i**). Scale bars, 30 mm.

sensing target $\xi$. Therefore, the sensing resolution $dp_1/d\xi$ can be written as

$$\frac{dp_1}{d\xi} = -\frac{p_0 v_{tank} + p_{atm} v_0}{(v_{tank} + v_1)^2} \cdot \frac{dv_1}{d\xi}, \qquad (2)$$

where, for example for the gripping test of the cylinders $\xi = D$, i.e., the diameter of the cylindrical objects in Fig. 4f and i. Moreover, $dv_1/d\xi$ represents the sensitivity of the internal geometric volume of the gripper to gripping cylindrical objects with different diameters.

Equation (2) shows that the variation of internal geometric volume when the soft actuator interacts with the environment in different ways causes the pressure change, which can be used to infer the interaction.

However, it is not trivial to compare the sensing resolutions of the soft actuators and grippers in Fig. 3 and Fig. 4, because these actuators vary in actuation pressure, internal volume and sensing targets. To give an example, since the initial internal geometric volumes $v_0$ of both commercial grippers in Fig. 4f and i are known, we can determine $v_1$ at equilibrium from equation (1) based on experimental measurements of $p_1$ (Table S1 and Fig. S11), from which we can then obtain $dp_1/dD$

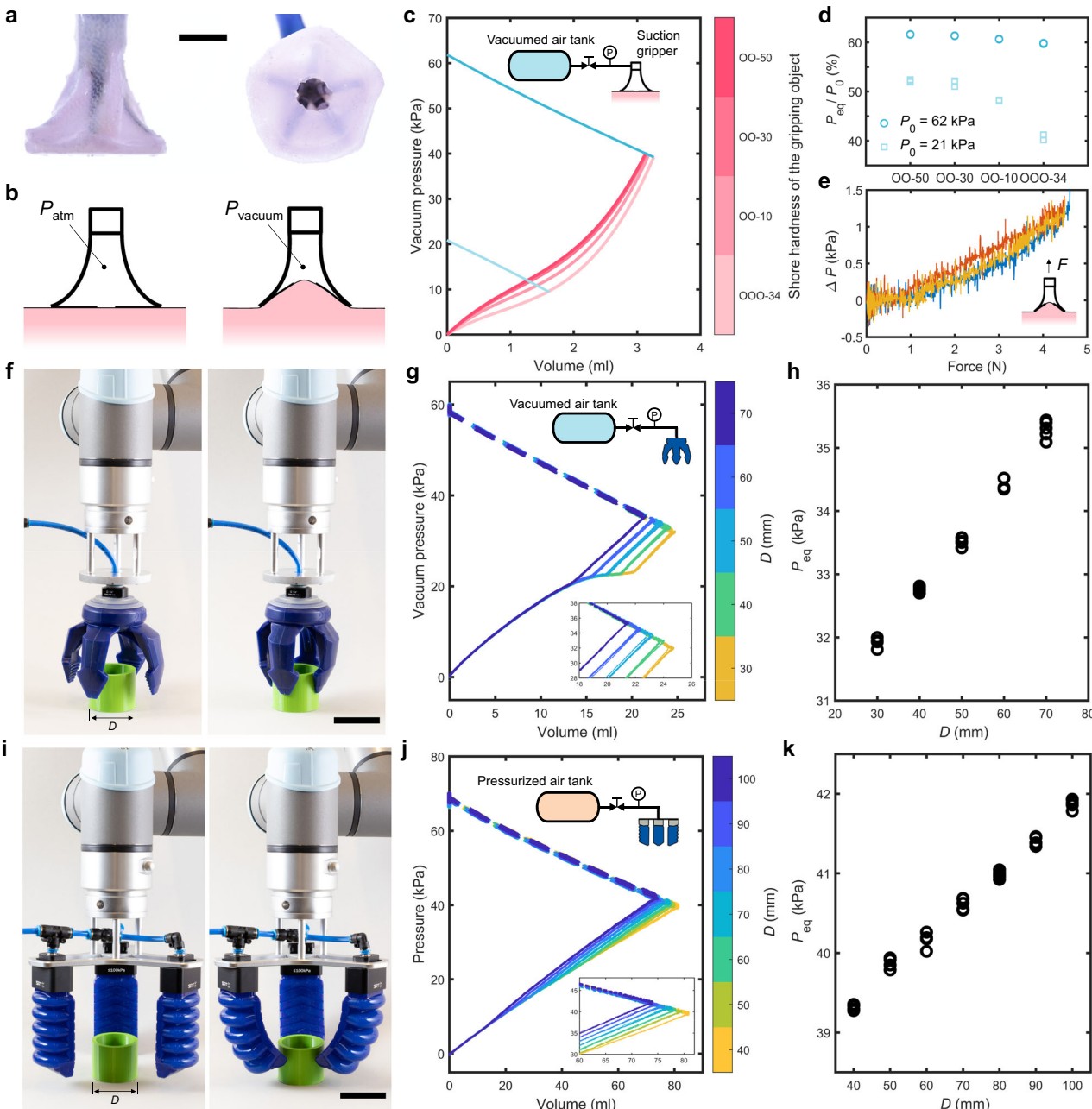

**Fig. 4 | Retrofitting the fluidic sensing approach to a suction cup (a-e) and two commercial soft grippers (f-k). a** Front and bottom views of the suction gripper. Scale bar, 10 mm. **b**, Schematic of the suction gripper attaching to a soft object. **c** Smoothened pressure-volume responses of the suction gripper (pink) and air tank (blue) when attaching to silicone samples with different shore moduli. **d** Experimental sensing results obtained using two different initial pressures $P_0$ in the air tank. **e** Force-pressure responses from three pulling tests on the suction gripper when attached to a silicone sample with a shore hardness of OO-30. Commercial vacuum (**f**) and pressurized (**i**) grippers gripping a cylindrical object with a diameter $D$. Scale bar, 50 mm. Corresponding pressure-volume relation for the soft gripper (solid) and tank (dashed) (**g**, **j**) and equilibrium pressure $P_{eq}$ in the system (**h**, **k**) for objects. Experimental results from five tests are shown for each $D$ (**g**, **h**, **j**, **k**).

according to equation (2). We find that the sensitivity of the internal geometric volume of the gripper to gripping different cylindrical objects equals $|dv_1/dD| = 0.16$ ml/mm for the vacuum gripper and $|dv_1/dD| = 0.13$ ml/mm for the PneuNet gripper, while the magnitude of the sensing resolution $|dp_1/dD| = 0.08$ kPa/mm of the vacuum gripper is twice that of the PneuNet gripper ($|dp_1/dD| = 0.04$ kPa/mm). According to equation (2), the smaller term $(v_{tank} + v_1)^2$ that is related to internal volumes in the case of the vacuum gripper contributes to the higher magnitude of sensing resolution when compared to the PneuNet gripper, even though the term $p_0 v_{tank} + p_{atm} v_0$ that is related to the total amount of air in the system is lower in the case of the vacuum gripper.

Equation (2) indicates that the sensing resolution is determined by the total amount of air $p_0 v_{tank} + p_{atm} v_0$ in the system, the internal geometric volume of the tank $v_{tank}$ and soft actuator $v_1$ at equilibrium, and the sensitivity of the internal geometric volume of the actuator to the sensing target $dv_1/d\xi$. The effects of system parameters, such as the stiffness and initial geometric volume of the soft actuator, on the sensing resolution depend on how these system parameters affect $p_0 v_{tank} + p_{atm} v_0$, $v_{tank}$, $v_1$ and $dv_1/d\xi$ in equation (2), which should be analyzed case by case. To give an example, we develop a basic model based on the interaction of an extension actuator (with a linear stiffness $k$) with a rigid wall in the Methods section Modeling the Fluidic Sensing Approach and show the effects of the linear actuator's initial

**Table 1 | Overview of the sensing performance of the actuators and grippers tested in this work**

| Actuator type | Sensing task | Absolute pressure change (kPa) | Maximum actuation pressure (kPa) | Relative pressure change to the maximum | Average sensing resolution |
|---|---|---|---|---|---|
| PneuNet actuator | Distance h between the actuator and a flat rigid plate from 0 to 40 mm | 1.7 | 26 | 6.5% | 0.04 kPa/mm |
| PneuNet gripper | Diameter D of cylindrical objects from 40 to 100 mm | 1.3 | 28.5 | 4.6% | 0.02 kPa/mm |
| Filament actuator | Angular displacement of the robotic arm from 0 degree to 60 degree | 1.8 | 163.7 | 1.1% | 1.72 kPa/rad |
| McKibben actuator | Angular displacement of the robotic arm from 0 degree to 30 degree | 5.9 | 32.7 | 18.0% | 11.27 kPa/rad |
| TPU bending actuator | Diameter D of cylindrical objects from 60 to 100 mm | 3 | 193.7 | 1.5% | 0.08 kPa/mm |
| Suction cup | Shore hardness of gripping objects from OOO-34 to OO-50 | 2.4 | 11 | 21.8% | 0.04 kPa/kPa |
| Commercial vacuum gripper | Diameter D of cylindrical objects from 30 to 70 mm | 3.4 | 35.3 | 9.6% | 0.09 kPa/mm |
| Commercial PneuNet gripper | Diameter D of cylindrical objects from 40 to 100 mm | 2.6 | 41.9 | 6.2% | 0.04 kPa/mm |

The modulus (at 100% strain) of Ecoflex OO-50 is 82.7 kPa according to the product sheet. The modulus (at 100% strain) of OOO-34 is estimated from that of OOO-35 (17 kPa) from the reference[63].

length, cross section, stiffness and initial tank pressure on the sensing resolution in Fig. S12. For example, for the modeled linear extension actuator, an increase in length (increase in initial volume) reduces the sensing resolution, while an increase in area (also an increase in initial volume) first increases and then decreases the sensing resolution. Despite the simplifications made in the model, we believe that it provides a framework for choosing available parameters for improving the sensing resolution.

Finally, it should be noted that the overall sensing accuracy is determined by both the sensing resolution of the actuator or gripper and the sensing accuracy of the pressure sensor used, and luckily, there are ample (relatively cheap) pressure sensors on the market that span various pressure ranges with sufficient accuracy for our purpose (Table S2). For example, in the tests with the PneuNet actuator in Fig. 1b, we used a ± 34.5kPa pressure sensor with an accuracy of ± 0.25% (of Full Scale Span). By dividing the pressure sensor error (± 0.1725kPa) by the average sensing resolution (0.04kPa/mm), we can obtain an overall sensing accuracy of ± 4.31mm for the PneuNet actuator. This sensing accuracy is valid for one-time measurement with the pressure sensor. In this work, however, we always do the pressure measurement over a period of time (which we were able to reduce to 0.05 seconds in Fig. S20) and calculate the average value, which gives a higher overall sensing accuracy, e.g., ± 1.7mm with a 95% confidence interval (Fig. S3a) for the PneuNet actuator.

## Closed-loop control with fluidic sensing

With the insights gained on sensing performance, we finally demonstrate that the fluidic sensing strategy is robust for closed-loop control in three applications: size sorting, tomato picking and ripeness detection. In the first closed-loop control experiment (Fig. 5a, b and Supplementary Video 4), we used a modified insertion sort algorithm which inserts cylinders of different diameters one by one at the correct position in a sorted array based on the fluidic sensing feedback. Every time a new object is gripped, its size is measured via the feedback pressure in the gripper that is averaged over a fixed time window (red band in Fig. 5b) after actuation. When the feedback pressure is smaller than the pressures in the sorted array, the robotic arm directly moves the new object to the end of the queue, e.g., the first and last actuation cycles in Fig. 5b. Otherwise, the robotic arm moves objects in the sorted array to make the correct position available for the new object, e.g., the third and sixth actuation cycles in Fig. 5b. The closed-loop control makes it possible to successfully sort the four cylindrical objects with random input order (Fig. S13, Fig. S14, Fig. S15 and Supplementary Video 4). The fluidic sensing strategy also works for sorting random objects with irregular shape (Fig. S16, Fig. S17 and Supplementary Video 4). Note that no calibration curve is needed here since the comparison between the equilibrium pressure values is sufficient for sorting.

In the second closed-loop control experiment (Fig. 5c, d and Supplementary Video 5), we perform a tomato picking experiment. We artificially increase the pressure in every step to demonstrate that the proposed sensing approach can provide feedback for picking automation. Note that in a more realistic setting, one would have likely used the highest pressure immediately. During each picking cycle, the algorithm compares the equilibrium pressure $P_{eq}$ in the gripper to a corresponding reference measurement $P_{ref}$ in free space, and evaluates $\Delta P = P_{eq} - P_{ref}$ to determine a successful or unsuccessful picking. When $\Delta P$ is smaller than a threshold, the algorithm regards it as an unsuccessful picking as the actuators are not deformed by the tomato. It then starts the next picking cycle with a higher actuation pressure. Once $\Delta P$ is larger than a threshold (0.2 kPa), the algorithm regards it as a successful picking. The gripper places the tomato on the table and moves to the next tomato in line. Note that the slip of the tomato out of the gripper in an unsuccessful picking can also be detected from the abrupt decrease in the pressure-time curve, which could potentially

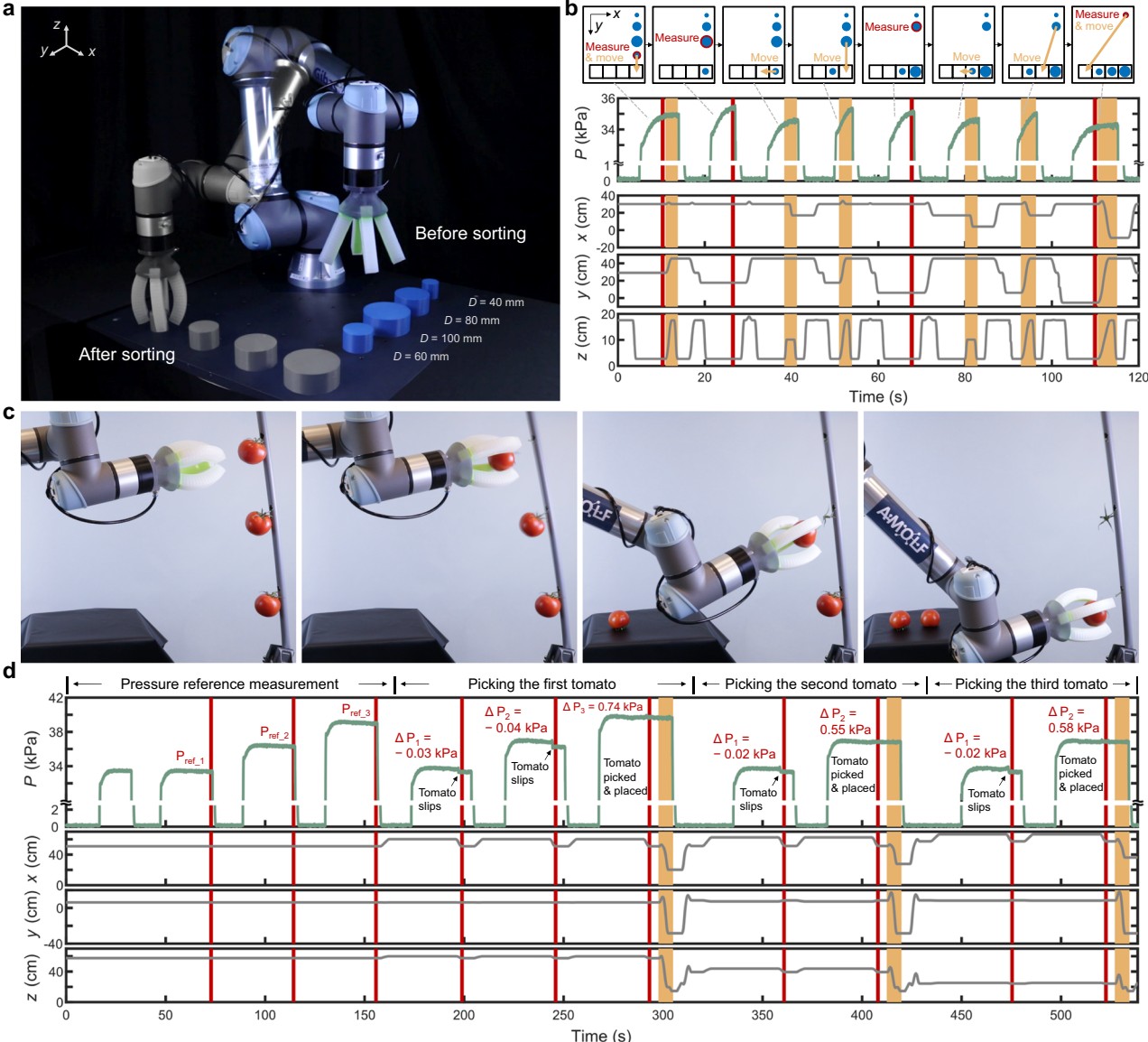

**Fig. 5 | Closed-loop control with fluidic sensing. a** Superimposed pictures taken at the start and end of the size sorting experiment. The input order of the cylindrical objects was randomly selected. **b** gripper actuation pressure and coordinates of the Tool Center Point (TCP) over time during the sorting. The manipulations during each actuation cycle are explained in the corresponding schematics. The red band in the plot represents the pressure feedback measurement, and the yellow band represents the object movement. The four pressure feedback measurements (average ± standard deviation) are $34.80 \pm 0.07$ kPa, $35.36 \pm 0.07$ kPa, $35.05 \pm 0.08$ kPa, $34.13 \pm 0.07$ kPa, respectively. **c** Snapshots of the tomato picking experiment representing the measurement of pressure reference $P_{ref}$ and picking of the three tomatoes, respectively. **d** Gripper actuation pressure and TCP coordinates over time during the tomato picking. The red band represents the measurement of equilibrium pressure $P_{eq}$, and the yellow band represents the tomato placement on the table. A pressure difference $\Delta P = P_{eq} - P_{ref}$ larger than 0.2 kPa is considered as a successful picking event.

provide additional sensing feedback. We tested a total of nine tomatoes in three runs, out of which six tomatoes were successfully picked and placed, one tomato was not picked by the gripper, the other two were successfully picked but not recognized because the tomato slipped into the palm of the gripper after being picked from the stem, resulting in a $\Delta P$ smaller than the threshold (Fig. S18 and Supplementary Video 5). As our soft gripper was not specifically designed for picking tomatoes, the design of the gripper should be optimized for this task. Importantly, this would not affect the retrofit implementation of our sensing approach.

In the third closed-loop control experiment (Fig. 6 and Supplementary Video 6), we detect the ripeness of tomatoes by applying the method mentioned in Fig. 2f and g to estimate the indentation depth. We demonstrate that the proposed sensing approach can provide feedback for automated sorting of an

overripe tomato from ripe tomatoes. For versatility, we select the commercial vacuum gripper for this demonstration, also as it has the largest sensing resolution (Table 1). The demonstration includes one cycle of calibration and five separate cycles of sensing to determine repeatability. The positions of the four tomatoes (including one overripe tomato) and one dummy are shuffled randomly between sensing cycles. All size predictions are based on one calibration process with 3D-printed rigid dummies, where the gripper pressure $P_{gripper}$ is compared to a reference response $P_{ref}$ to determine $\Delta P = P_{gripper} - P_{ref}$ over time for different diameters of the object that is being gripped (Fig. 6b and c). As explained by the stiffness sensing method in Fig. 2f and g, the size of the tomato upon gripping can be inferred by the time of first contact $t_c$ (Fig. 6d), and the size at equilibrium can be inferred by the $\Delta P$ at equilibrium (Fig. 6e).

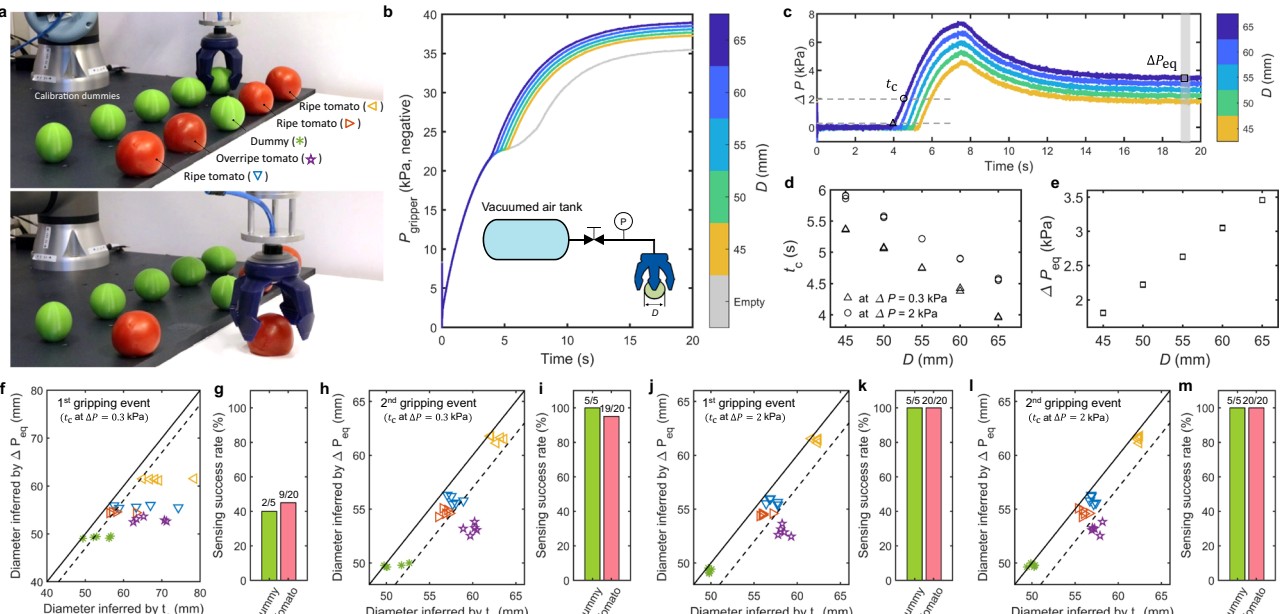

**Fig. 6 | Picking out an overripe tomato with fluidic sensing. a** Snapshots of the closed-loop control demonstration. Five 3D-printed rigid dummies (*D* = 45, 50, 55, 60, 65 mm) are used for calibration. The other five objects are tested for sensing, including three ripe tomatoes, one overripe tomato and one dummy with a diameter *D* = 50 mm. The whole demonstration includes one cycle of calibration and five cycles of sensing. In each sensing cycle, two gripping events are carried out for each object. The positions of the objects are shuffled randomly between cycles. **b-e** Calibration results. The pressure response $P_{gripper}$ of the gripper when gripping a calibration dummy is compared to the pressure response $P_{ref}$ of the gripper when gripping nothing, to obtain the pressure difference $\Delta P = P_{gripper} - P_{ref}$. The time of

first contact $t_c$ and the pressure difference at equilibrium $\Delta P_{eq}$ can be used to infer the object diameter upon gripping (**d**) and at equilibrium (**e**), respectively. Experimental results from three measurements are plotted for each *D* (**b–e**). **f–m** Sensing results. The solid line in **f, h, j, l** represents that the object diameter inferred by $t_c$ equals that inferred by $\Delta P_{eq}$, the dashed line represents the object diameter inferred by $\Delta P_{eq}$ is 3 mm smaller than that inferred by $t_c$. For the ripe tomatoes and rigid dummy, a measurement above the dashed line is considered as a successful sensing event. Conversely, for the overripe tomato, a measurement below the dashed line indicates sensing success.

It is important to note that the method that uses the time of first contact is strongly affected by the alignment of the tomato inside the gripper, leading to an early rise of $\Delta P$-time curve and inaccurate size predictions of the tomato upon gripping and sensing success rates of 40% and 45% (Fig. 6f and g) for the rigid dummy and tomatoes, respectively. To increase the sensing success rate, we can perform the gripping event twice, the first to center the object inside the gripper, and the second gripping event to extract sensing feedback (Fig. 6h and i). Alternatively, we can choose $t_c$ at higher $\Delta P$ values for both calibration and sensing (Fig. 6j and k), so that the object has been effectively centered during sensing. With $t_c$ at $\Delta P$ = 2kPa, sensing during either the first or second gripping event gives 100% success rates for both rigid dummy and tomatoes (Fig. 6j-m). We can also infer the initial size of the tomato by $\Delta P_{max}$ (Fig. S19 and Supplementary Video 6) instead of $t_c$ to avoid the influence of misalignment, which gives 100% and 90% success rate for rigid dummy and tomatoes, respectively. While we were able to pick out the rotten tomato consistently, since the gripper squeezes tomatoes for ripeness detection in this method, post-harvest studies should be performed in the future to avoid extra damage to the produce when applying the method in practical applications.

It should be noted that all the closed-loop control demonstrations above were performed under quasistatic conditions. We tested the sensing strategy at different actuation speeds (Fig. S20) and find that the effectiveness of the sensing strategy is not affected by the actuation speed, as long as the sensing feedback is collected after the actuator comes into contact with the environment. To speed up the sensing process for real-world applications, it is important to ensure an actuation speed that is high enough for the interaction with the environment to happen before the collection of fluidic sensing feedback. To prove the feasibility, we successfully

implemented fast sensing in the size sorting demonstration, where pressure values are collected 0.5 s (instead of 5 s) after the opening of the valve between the air tank and gripper (Fig. S21 and Supplementary Video 4).

## Discussion

In conclusion, we present a versatile fluidic sensing strategy that relies on measuring the fluidic input response instead of embedded sensing elements into the soft actuators. The soft robot-environment interaction can be accurately interpreted with one pressure sensor that is connected remotely. We show that the proposed strategy can be retrofitted to a wide range of soft devices, and implemented in closed-loop control of gripping applications. We believe that this relatively straightforward integration of sensing capabilities makes it readily available for other soft robotic devices and applications, including wearable assistive devices[53] and soft locomotive robots[48,54], without the need to alter the design of the soft device itself.

While we were able to retrofit our sensing approach to a range of soft actuators and grippers, it should be noted that the air tank method in Fig. 1b is not directly applicable to soft actuators driven by incompressible liquid, as the method depends on the compressibility of air and the final pressure balance between the tank and actuator. This could be solved by using a flexible tank (e.g., a balloon), such that the compressibility of the air is replaced by the elasticity of the tank. A simpler approach could instead be to use the pressure control (Fig. 1c) or flow control (Fig. 1d) method to obtain sensory feedback from the liquid-driven actuator's interaction with the environment.

According to equation (2), the sensing resolution is influenced by the final internal volume $v_1$ of the actuator after the interaction with the

environment and its derivative over the sensing target $dv_1/d\xi$. Yet, there is no trivial relationship between the initial volume of the actuator and the sensing resolution. Even though we demonstrate that we can retrofit sensing to pneumatic actuators, optimizing the sensing resolution by for example changing the tank size and the initial tank pressure should be done on a case by case basis, and can best be done by experimentally obtaining the relation between pressure and volume for specific interactions with the environment. For example, if we want to apply our strategy to pneumatic actuators that can generate complex motions with multiple degrees of freedom[55], the sensing resolution depends on how the interaction of the actuator with the environment affects $v_1$ and $dv_1/d\xi$ in equation (2), which is not straightforward to predict beforehand and depends on the application.

Moreover, we did find relatively small variations over time during cyclic gripping. These variations may be due to the performance of the soft actuator, pressure regulator and pressure sensor, or environment variations like the temperature. Comparing the sensing response to a reference helps reduce the influence of long-term system and environment variations, as also demonstrated in the closed-loop control examples. Still, any non-unique pressure-volume response of a soft actuator would cause inaccurate sensing. Moreover, since the proposed sensing principle uses the soft actuator itself as a sensor, any fragility or unreliability of the soft actuator would have a direct influence on the sensing performance. Additionally, the soft actuation of more complex devices should be designed such that it achieves sensitivity to the sensing task.

Moving forward, machine learning can potentially be applied to read the higher-order difference in the pressure-time curves of the soft robot for various interactions with the environment to achieve more complex sensing applications[12,56]. Although our sensing strategy removes the need of embedding or attaching sensors to the soft actuator, the hardware of the system is still bulky and may not be suitable (yet) for small-scale untethered soft robotic system[57]. To further reduce the bulkiness of the system, the fluidic sensory feedback from the soft robot-environment interaction can be potentially read out by soft pneumatic valves[58–60] to build electronic-free soft robots[59,61,62] that can sense and respond to their environment. The basic yet powerful principles studied in this work make it possible to bring (some) sensing capabilities to most soft fluidic devices without the need for design changes, and paves the way towards new functionalities in soft interactive devices and systems for real world applications.

## Methods
Details on the methods are provided in Supplementary Information.

## Data availability
The experimental and numerical data that support the findings of this work and computer algorithms necessary for running the analysis have been deposited at Zenodo (https://doi.org/10.5281/zenodo.10276372).

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

## Acknowledgements

We are grateful to Niels Commandeur for technical support. This work is part of the Dutch Research Council (NWO), 4TU HTSF Soft Robotics Programme, and supported by European Research Council Starting Grants (grant agreement ID: 948132).

## Author contributions

S.Z. and J.T.B.O. conceived the main concepts. S.Z., S.P., J.d.V., V.G.K., A.S. and J.T.B.O. designed and performed the experiments, and analyzed data. S.Z. and J.T.B.O. wrote the manuscript with contributions from all authors.

## Competing interests

The authors declare no competing interests.

## Additional information

**Peer review information** : Nature Communications thanks Gursel Alici, Ahmad Rafsanjani and the other, anonymous, reviewer for their contribution to the peer review of this work. A peer review file is available.

