## [Peer Review File · Nature Communications]

REVIEWER COMMENTS

Reviewer #1 (Remarks to the Author):

This paper presents a strategy for measuring interactions between a soft actuator and the environment via the pressure differential within the actuator itself. The paper is well written and the method is demonstrated using a series of off-the-shelf fluidic actuators.

Measuring pressure is not new (references 21-32), nor is the idea to do this outside the actual actuator (23, 27). Albeit 23 and 27 both integrate additional sensors to measure strain, many of the same conclusions have already been drawn. In particular, both

[23] Farrow, N. and Correll, N., 2015, September. A soft pneumatic actuator that can sense grasp and touch. In 2015 IEEE/RSJ International Conference on Intelligent Robots and Systems (IROS) (pp. 2317-2323). IEEE.

[27] Hughes, J., Spielberg, A., Chounlakone, M., Chang, G., Matusik, W. and Rus, D., 2020. A simple, inexpensive, wearable glove with hybrid resistive-pressure sensors for computational sensing, proprioception, and task identification. *Advanced Intelligent Systems*, 2(6), p.2000002.

perform similar experiments distinguishing the width of objects or showing pressure signatures that arise when holding multiple different objects. (The full references are reproduced here as the paper does not mention the title of 23.)

As this information is very noisy, however, the method is unlikely to be used on its own, in particular if other sensing modalities are available. More importantly, the paper under consideration does not provide any additional modeling above 23 and 27, nor does it offer additional quantitative result (e.g. a systematic study of distinguishing different objects in a statistically significant way).

Reviewer #2 (Remarks to the Author):

It is a solid piece of work, supported by a comprehensive amount of experimental work to demonstrate the efficacy of the fluidic sensing in soft robotics. The following comments and suggestions are offered to substantiate its contribution and significance.

1. As explicitly stated in the manuscript, the idea of the fluidic sensing is not new [34], but is revisited in this study to “gain a better understanding of the underlying principles that allow for fluidic self-sensing, and determine if we can infer the interaction of a wide variety of soft actuator with their environment by measuring and analyzing the fluidic response of the enclosed cavity”. The honest and clear description of the literature is highly appreciated. This study takes this concept to a next level to demonstrate its validity in soft gripping/robotic applications. No previous soft robotic study investigated into the fluidic sensing concept, to the best of my knowledge.

2. How valid are the results in Figure 1, for the actuators made of different materials, especially from a flexible TPU requiring a much higher pressure to bend actuators like PeuNet, e.g., <https://doi.org/10.1089/soro.2016.0030>.

3. Please discuss how the sensing/calibration curves change for the actuators requiring relatively higher input pressures? As stated for the filament actuator, less sensing resolution will be achieved for these actuators.

4. While it is appreciated to apply this concept to different actuators, there is no indication of the actuators typically requiring higher pressures (>250 kPa). These actuators will typically show smaller differences in their pressure-volume curves. Yes, the filament actuator requires pressures at this level. However, the volume of these actuators is much smaller to result in enough change in the pressure-volume curves. My point is more for actuators with a larger volume and requiring a high pressure in the range of >250 kPa.

5. Although the effect of the internal volume of the air tank on the tank’s pressure-volume response is being discussed or remarked, how about the internal volume of the PneuNet like actuators? Also, how about PneuNet like actuators with a 3D movement such as bending and twisting motions simultaneously, (e.g., <https://doi.org/10.1089/soro.2019.0015>) as opposed to 2D bending movement of PneuNet actuators?

6. When picking up a tomato or similar with more softness (especially softer than the gripper), how will the closed-loop control approach will work? Reaching a larger threshold may damage the produce softer object or agricultural produce.

Reviewer #3 (Remarks to the Author):

A Retrofit Sensing Strategy for Soft Fluidic Robots

By S. Zou et al. (submitted to Nature Communications)

This work introduces an innovative self-sensing strategy for soft fluidic actuators, utilizing the measurement of internal pressure variations within actuator cavities. These pressure changes are responsive to interactions with the environment, such as contact with surfaces of different textures or objects with varying dimensions. The key advantage of this approach lies in its simplicity, as it eliminates the need for sensor redesign or compromises in compliance due to the integration of rigid sensing components. The presented method demonstrates versatility by showcasing its applicability to a wide range of fluidic actuators, including commercially available ones. The paper incorporates precision experiments with meticulously controlled parameters, covering diverse actuation and sensing protocols. For instance, it investigates scenarios where a soft actuator makes contact with the ground at different heights or interacts with regular geometrical objects (e.g., blocks, cylinders) of varying dimensions. By comparing the pressure difference changes to a reference configuration (freely moving actuator), the authors utilize this feedback to derive tactile information. The work demonstrates comprehensiveness, with precise experimental execution and compelling demonstrations that substantiate the proposed concept.

However, there are certain considerations to be addressed before the publication of this work in Nature Communications. The foremost concern is the relatively low range of pressure change observed during robot-environment interactions (~ 1 kPa). This necessitates the use of highly accurate pressure sensors and may overlap with naturally occurring pressure variations in real-world scenarios.

To improve the paper, I suggest addressing the following comments:

1. The authors have based their fluidic sensing on the ideal gas law, but it would be beneficial to discuss whether similar pressure fluctuations could occur in liquid-driven actuators, where the fluid is incompressible (e.g., with water). Clarifying the applicability of this method to liquid-driven actuators would enhance the paper's completeness.

2. While various actuators were examined in this work, the pressure difference observed in most cases was approximately 1 kPa. It would be insightful to investigate the effect of actuator stiffness on the sensitivity of the proposed technique. This analysis would provide valuable insights into the relationship between actuator properties and the sensing strategy's effectiveness.

3. Although the experiments conducted in this study were primarily under quasistatic conditions, it is important to consider real-world applications where actuation occurs at higher speeds (e.g., Soft Robotics Inc. pick-and-place demonstrations). It would be valuable to evaluate the performance of the sensing strategy at different actuation rates, particularly for simple cases. This analysis would further demonstrate the robustness of the proposed approach.

4. In the case of commercial actuators, it is notable that the effect of pressure difference is more pronounced in vacuum-driven soft grippers compared to positive pressure actuators (Fig. 3m vs Fig. 3p). It would be beneficial to provide insights into this discrepancy and explore the underlying factors contributing to the varying response between these two types of actuators.

5. The closed-loop control in this work is based on a simple pressure difference thresholding. Considering the potential of the proposed self-sensing technique, it would be interesting to investigate whether richer information beyond pressure change can be extracted. Expanding on the capabilities of the sensing strategy to gather more comprehensive data would add depth to the study.

6. The tomato picking demonstration showcased in the paper is intriguing. Given that the actuator can discern surfaces with different stiffness (e.g., Ecoflex vs. rigid surfaces), it would be compelling to explore the application of this method in sorting ripened from unripened tomatoes. This extension of the work would demonstrate its practicality in agricultural automation and highlight its potential for addressing real-world challenges.

7. If applicable, please report the success rate of the tomato picking experiment. Including this information would provide valuable insights into the performance of the sensing strategy in a specific application scenario.

By addressing these comments and further expanding the paper's scope, this work has the potential to make a significant contribution to the field of self-sensing in fluidic actuators.

Response to the Reviewers' Comments for Manuscript: "A Retrofit Sensing Strategy for Soft Fluidic Robots"

In the following, we address each of the Reviewers' comments/suggestions (*in italic*). The revisions are highlighted in blue.

Response to Reviewer #1

Comment 1A

This paper presents a strategy for measuring interactions between a soft actuator and the environment via the pressure differential within the actuator itself. The paper is well written and the method is demonstrated using a series of off-the-shelf fluidic actuators.

Measuring pressure is not new (references 21-32), nor is the idea to do this outside the actual actuator (23, 27). Albeit 23 and 27 both integrate additional sensors to measure strain, many of the same conclusions have already been drawn. In particular, both

[23] Farrow, N. and Correll, N., 2015, September. A soft pneumatic actuator that can sense grasp and touch. In 2015 IEEE/RSJ International Conference on Intelligent Robots and Systems (IROS) (pp. 2317-2323). IEEE.

[27] Hughes, J., Spielberg, A., Chounlakone, M., Chang, G., Matusik, W. and Rus, D., 2020. A simple, inexpensive, wearable glove with hybrid resistive-pressure sensors for computational sensing, proprioception, and task identification. Advanced Intelligent Systems, 2(6), p.2000002.

perform similar experiments distinguishing the width of objects or showing pressure signatures that arise when holding multiple different objects. (The full references are reproduced here as the paper does not mention the title of 23.)

We agree with the Reviewer that fluidic pressure sensing has been explored in the literature (references [21] to [32]), as also pointed out in our manuscript. The main contribution of our work is that we reveal that the change in pressure-volume response of a soft fluidic actuator during activation can be used to infer interactions with the environment, and in particular that this sensing principle can be applied to a wide range of soft fluidic actuators without the need for design changes. This contribution has also been pointed out by other Reviewers (Reviewer #2: *As explicitly stated in the manuscript, the idea of the fluidic sensing is not new ... This study takes this concept to a next level to demonstrate its validity in soft gripping/robotic applications.* Reviewer #3: *The key advantage of this approach lies in its simplicity, as it eliminates the need for sensor redesign or compromises in compliance due to the integration of rigid sensing components.*).

In comparison to our main contribution that focuses on retrofitting sensing capabilities without locally incorporating sensors, the sensing approaches used in [23] and [27] still require a redesign of the actuator to incorporate the sensing capabilities, and to integrate the sensor with the actuator. Therefore, these sensing approaches cannot directly be applied to new type of soft fluidic actuators, without redesign.

Specifically, reference [23] proposed a sensing method with integrated strain and pressure sensors for soft pneumatic actuators. Note that the sensing method in [23] requires both the strain and pressure sensors to distinguish successful grasps of cylindrical objects with varying diameter. To apply this method to a new type of soft fluidic actuator, such as the filament actuator in references [46-48], the strain sensor in [23] needs to be redesigned to be integrated on the filament actuator and the performance of the filament actuator will also be influenced by the integrated sensor. Additionally, reference [27] proposes a fluidic tube sensor to measure the interactions between wearable devices and the environment. The sensing principle in [27] requires a fixed amount of fluid in the tube sensor, which therefore cannot act as an actuator. Following the sensing method in [27], one could attach the fluidic tube sensor to another soft fluidic actuator, such as the work in reference [28]. However, sensor/actuator redesign is still necessary when applying the sensing method in [27] to other soft fluidic actuators and the integration will inevitably affect the performance of the actuator.

Finally, it seems that during compiling of our article in latex something went wrong, and all titles of the conference papers that we wanted to refer to in the original manuscript, including [23], were missing due to the citation package we used. We have removed this error, so that these references (including [23]) now also appear in the article.

Comment 1B

As this information is very noisy, however, the method is unlikely to be used on its own, in particular if other sensing modalities are available. More importantly, the paper under consideration does not provide any additional modeling above 23 and 27, nor does it offer additional quantitative result (e.g. a systematic study of distinguishing different objects in a statistically significant way).

While the pressure sensing information is often considered to be noisy and not useful in practical applications, we demonstrate in our work the wide applicability of the fluidic sensing strategy in various sensing scenarios and robustness in practical applications with closed-loop control, such as size sorting and tomato picking. Additionally, we added results on tomato ripeness detection (see the response to Comment 2F), and we were also able to considerably reduce the time needed for the sensing (see the response to Comment 3E).

As our main contribution in the original version of the manuscript was to show that the fluidic sensing strategy can be easily implemented on a wide range of soft fluidic actuators without the need for modeling, we did not include any modeling information in the original manuscript. Instead, we focused on the experimental characterization of the response of different kinds of soft actuators and grippers to external influences (in comparison to most papers that focus on a single type of actuators).

However, given that all Reviewers would welcome additional modeling results, we realize that additional modeling could provide more insights into the influence of system parameters on the sensing performance, for example to optimize sensing resolution. Therefore, we added two parts to the manuscript. In the first part we focus on characterizing the sensing resolution based on experimen-

tal data, in order to be able to make a comparison between the performance of the different kinds of soft actuators and grippers. In the second part, we derive a model that demonstrates the effect of certain parameters on the sensing capabilities.

We added the first part on sensing resolution to a new section **Characterizing the Sensing Resolution** in the main manuscript:

Actuator type	Sensing task	Absolute pressure change (kPa)	Maximum actuation pressure (kPa)	Relative pressure change to the maximum (kPa)	Average sensing resolution
PneuNet actuator	Distance h between the actuator and a flat rigid plate from 0 to 40 mm	1.7	26	6.5%	0.04 kPa/mm
Suction cup	Shore hardness of gripping objects from OOO-34 to OO-50	2.4	11	21.8%	0.04 kPa/kPa
TPU bending actuator	Diameter D of cylindrical objects from 60 to 100 mm	3	193.7	1.5%	0.08 kPa/mm
Filament actuator	Angular displacement of the robotic arm from 0 degree to 60 degree	1.8	163.7	1.1%	1.72 kPa/rad
McKibben actuator	Angular displacement of the robotic arm from 0 degree to 30 degree	5.9	32.7	18.0%	11.27 kPa/rad
PneuNet gripper	Diameter D of cylindrical objects from 40 to 100 mm	1.3	28.5	4.6%	0.02 kPa/mm
Commercial vacuum gripper	Diameter D of cylindrical objects from 30 to 70 mm	3.4	35.3	9.6%	0.09 kPa/mm
Commercial PneuNet gripper	Diameter D of cylindrical objects from 40 to 100 mm	2.6	41.9	6.2%	0.04 kPa/mm

Table 1: **Overview of the sensing performance of the actuators and grippers tested in this work.** The modulus (at 100% strain) of Ecoflex OO-50 is 82.7 kPa according to the product sheet. The modulus (at 100% strain) of OOO-34 is estimated from that of OOO-35 (17 kPa) from the reference (53).

“Fig. 3 and Fig. 4 provide a general picture of how the variations in pressure-volume curves between the soft actuators and grippers lead to different sets of equilibrium points based on the conservation of air mass. The absolute pressure change observed during robot-environment interactions ranges from 1.3 kPa to 5.9 kPa among the soft actuators and grippers we tested (Table 1). Even though the relative pressure difference (pressure difference due to interaction with the environment in comparison to maximum pressure obtained in the actuator) might be smaller for actuators that require higher inflation pressure (e.g., TPU and filament actuators), the absolute pressure change for all actuators we tested is in the same order of magnitude (Table 1). Note that the average sensing resolution can be determined by dividing the absolute pressure change by the tested range of sensing target and is therefore not affected by a lower relative pressure difference.

In order to find out the underlying factors that determine the sensing resolutions of the soft actuators and grippers, we consider both the initial and final states of the system. At the initial state, the air tank with an internal geometric volume v_{tank} is pressurized at $p_{\text{tank}} = p_0$, and the actuator with an internal geometric volume $v_{\text{act}} = v_0$ is at atmosphere pressure $p_{\text{act}} = p_{\text{atm}}$. At the final state, the air tank and the actuator reach the same pressure $p_{\text{tank}} = p_{\text{act}} = p_1$, and the internal geometric volume of the actuator becomes $v_{\text{act}} = v_1$. Assuming constant temperature, since the total amount of air mass inside the system (air tank and the actuator) stays constant, according to the ideal gas law, we have

$$p_0 v_{\text{tank}} + p_{\text{atm}} v_0 = p_1 v_{\text{tank}} + p_1 v_1. \quad (1)$$

Note that the absolute pressure here is indicated in lowercase letter to distinguish it from the relative pressure (with respect to atmospheric pressure) that is used elsewhere in the manuscript. When the total amount of air inside the system remains constant, v_1 only depends on the interaction of the soft actuator with the environment, i.e., the sensing target ξ . Therefore, the sensing resolution $dp_1/d\xi$ can be written as

$$\frac{dp_1}{d\xi} = -\frac{p_0 v_{\text{tank}} + p_{\text{atm}} v_0}{(v_{\text{tank}} + v_1)^2} \cdot \frac{dv_1}{d\xi}, \quad (2)$$

where, for example for the gripping test of the cylinders $\xi = D$, i.e., the diameter of the cylindrical objects in Fig. 4f and i. Moreover, $dv_1/d\xi$ represents the sensitivity of the internal geometric volume of the gripper to gripping cylindrical objects with different diameters. Equation 2 shows that essentially, the variation of internal geometric volume when the soft actuator interacts with the environment in different ways causes the pressure change, which can be used to infer the interaction.

However, it is not trivial to compare the sensing resolutions of the soft actuators and grippers in Fig. 3 and Fig. 4, because these actuators vary in actuation pressure, internal volume and sensing targets. To give an example, since the initial internal geometric volumes v_0 of both commercial grippers in Fig. 4f and i are known, we can determine v_1 at equilibrium from equation 1 based on experimental measurements of p_1 (Table S1 and Fig. S11), from which we can then obtain dp_1/dD according to equation 2. We find that the sensitivity of the internal geometric volume of the gripper to gripping different cylindrical objects equals $|dv_1/dD| = 0.16$ ml/mm for the vacuum gripper and $|dv_1/dD| = 0.13$ ml/mm for the PneuNet gripper, while the magnitude of the sensing resolution $|dp_1/dD| = 0.08$ kPa/mm of the vacuum gripper is twice that of the PneuNet gripper ($|dp_1/dD| = 0.04$ kPa/mm). According to equation 2, the smaller term $(v_{\text{tank}} + v_1)^2$ that is related to internal volumes in the case of the vacuum gripper contributes to

Figure S11: **Internal volume estimation of the commercial grippers when gripping cylindrical objects with different diameters.** The dashed lines in **c** and **f** represent linear fits of the data.

	Initial tank pressure p_0 (kPa, absolute)	Tank internal Volume v_{tank} (ml)	Initial gripper pressure p_{atm} (kPa, absolute)	Initial gripper internal volume v_0 (ml)	Equilibrium pressure p_1 (kPa, absolute)	Equilibrium gripper internal volume v_1 (ml)	dv_1/dD (ml/mm)	Calculated dp_1/dD (kPa/mm)	Measured dp_1/dD (kPa/mm)
Piab gripper, $D = 60$ mm	41.55	100	101.325	47.9	66.9	34.79	0.16	-0.08	-0.08
SMT gripper, $D = 60$ mm	170.88	300	101.325	96.9	141.49	131.61	-0.13	0.04	0.04

Table S1: **Comparison of the two commercial grippers.** The dv_1/dD is obtained from the linear fits in Fig. S11c and f. The calculated dp_1/dD is based on equation 2. The measured dp_1/dD is obtained from the linear fits of the data in Fig. 4h and k.

the higher magnitude of sensing resolution when compared to the PneuNet gripper, even though the term $p_0v_{\text{tank}} + p_{\text{atm}}v_0$ that is related to the total amount of air in the system is lower in the case of the vacuum gripper.

Equation 2 indicates that the sensing resolution is determined by the total amount of air $p_0v_{\text{tank}} + p_{\text{atm}}v_0$ in the system, the internal geometric volume of the tank v_{tank} and soft actuator v_1 at equilibrium, and the sensitivity of the internal geometric volume of the actuator to the sensing target $dv_1/d\xi$. The effects of system parameters, such as the stiffness and initial geometric volume of the soft actuator, on the sensing resolution depend on how these system parameters affect $p_0v_{\text{tank}} + p_{\text{atm}}v_0$, v_{tank} , v_1 and $dv_1/d\xi$ in equation 2, which should be analyzed case by case. To give an example, we develop a basic model based on the interaction of an extension actuator (with a linear stiffness k) with a rigid wall in Methods section **Modeling the Fluidic Sensing Approach** and show the effects of the linear actuator’s initial length, cross section, stiffness and initial tank pressure on the sensing resolution in Fig. S12. For example, for the modeled linear extension actuator, an increase in length (increase in initial volume) reduces the sensing resolution, while an increase in area (also an increase in initial volume) first increases and then decreases the sensing resolution. Despite the simplifications made in the model, we believe that it provides a framework for choosing available parameters for improving the sensing resolution.

Finally, it should be noted that the overall sensing accuracy is determined by both the sensing resolution of the actuator or gripper and the sensing accuracy of the pressure sensor used, and luckily, there are ample (relatively cheap) pressure sensors on the market that span various pressure ranges with high enough accuracy for our purpose (Table S2). For example, in the tests with the PneuNet actuator in Fig. 1b, we used a ± 34.5 kPa pressure sensor with an accuracy of $\pm 0.25\%$ (of Full Scale Span). By dividing the pres-

Pressure sensor	Pressure range	Accuracy	Price (in Euros)
NPC-1220 series	34 to 689 kPa	$\pm 0.1\%$	44
Nidec P-2000 series	49 to 981 kPa	$\pm 0.12\%$	20-30
Honeywell ABP series	6 kPa to 1 MPa	$\pm 0.25\%$	20-30
Honeywell SSC series	160 Pa to 1 Mpa	$\pm 0.25\%$	50-80
NXP MPX5100 series	100 kPa	$\pm 2.5\%$	23

Table S2: **Comparison of selected commercial analog pressure sensors.** Prices are according to Mouser Electronics in 2023.

sure sensor error ($\pm 0.1725\text{kPa}$) by the average sensing resolution (0.04kPa/mm), we can obtain an overall sensing accuracy of $\pm 4.31\text{mm}$ for the PneuNet actuator. This sensing accuracy is valid for one-time measurement with the pressure sensor. In this work, however, we always do the pressure measurement over a period of time (which we were able to reduce to 0.05 seconds in Fig. S20) and calculate the average value, which gives a higher overall sensing accuracy, e.g., $\pm 1.7\text{mm}$ with a 95% confidence interval (Fig. S3a) for the PneuNet actuator.”

Moreover, we added the second part on a model that demonstrates the effect of certain parameters on the sensing resolution to the methods section of the manuscript:

“We develop a basic model based on the interaction of an extension actuator (with a linear stiffness k) with a rigid wall (Fig. S12), to characterize and better understand the dynamics between the air tank and actuator in the proposed sensing strategy. The linear actuator is initially at atmosphere pressure $p_{\text{act}} = p_{\text{atm}}$, while the tank with compressed air starts at $p_{\text{tank}} = p_0$. We take the sensing target as the initial distance L between the tip of the actuator and the wall. The valve between the air tank and actuator is opened at $t = 0$, so that the system will reach the equilibrium pressure p_1 . Assuming incompressible and laminar flow between the air tank and actuator, the Hagen-Poiseuille equation that characterizes the flow rate between air tank and actuator can be written as

$$\frac{dV}{dt} = \frac{p_{\text{tank}} - p_{\text{act}}}{r}, \quad (3)$$

where V represents the volume of air that transfers from the air tank to the actuator, r represents the flow resistance between the air tank and actuator, and p_{tank} and p_{act} indicate the absolute pressure in the air tank and actuator, respectively. According to the ideal gas law, we have

$$p_{\text{tank}}v_{\text{tank}} = n_{\text{tank}}RT, \quad (4)$$

and

$$p_{\text{act}}v_{\text{act}} = n_{\text{act}}RT, \quad (5)$$

where R represents the ideal gas constant, T represents the absolute temperature of the air, and v_{tank} and v_{act} represent the internal geometrical volume of the air tank and actuator, respectively. The volume v_{tank} is constant, while v_{act} depends on the interaction of the actuator with its environment. Assuming the internal volume of the actuator does not change anymore after the actuator comes into contact with the wall, we have

$$v_{\text{act}} = \begin{cases} A(l_0 + \frac{(p_{\text{act}} - p_{\text{atm}})A}{k}) & \text{if } \frac{(p_{\text{act}} - p_{\text{atm}})A}{k} < L, \\ A(l_0 + L) & \text{if } \frac{(p_{\text{act}} - p_{\text{atm}})A}{k} \geq L, \end{cases} \quad (6)$$

where A and l_0 represent the cross section and initial length of the actuator, respectively. By substituting equation 6 into equation 5, we have

$$p_{\text{act}} = \begin{cases} \frac{-(Al_0 - \frac{p_{\text{atm}}A^2}{k}) + \sqrt{(Al_0 - \frac{p_{\text{atm}}A^2}{k})^2 + 4n_{\text{act}}RT\frac{A^2}{k}}}{\frac{2A^2}{k}} & \text{if } \frac{(p_{\text{act}} - p_{\text{atm}})A}{k} < L, \\ \frac{n_{\text{act}}RT}{A(l_0 + L)} & \text{if } \frac{(p_{\text{act}} - p_{\text{atm}})A}{k} \geq L. \end{cases} \quad (7)$$

Parameter	Value	Unit
M , molar mass of air	0.02896	kg/mol
ρ , density of air	1.2922	kg/m ³
R , ideal gas constant	8.314	J/(mol·K)
T , temperature of the air	273.15	K
v_{tank} , internal geometric volume of the air tank	0.0001	m ³
p_0 , initial tank pressure, absolute	170000	Pas
r , flow resistance	5.7139E+08	pas·s/m ³
l_0 , initial length of the actuator	0.05	m
A , cross section of the actuator	0.0004	m ²
p_{atm} , atmosphere pressure	101325	Pas
k , actuator stiffness	400	N/m

Table S3: **Parameter values used in the fluidic sensing model.**

Combining equations 3, 4 and 5, we obtain two differential equations that describe the change of the amount of air mass n_{tank} in the air tank and n_{act} in the actuator over time

$$\frac{Mdn_{\text{tank}}}{\rho dt} = -\frac{\frac{n_{\text{tank}}RT}{v_{\text{tank}}} - p_{\text{act}}}{r}, \quad (8)$$

$$\frac{Mdn_{\text{act}}}{\rho dt} = \frac{\frac{n_{\text{tank}}RT}{v_{\text{tank}}} - p_{\text{act}}}{r}, \quad (9)$$

where M and ρ represent the molar mass and density of air, respectively, and p_{act} is given by equation 7. With initial conditions $n_{\text{tank}}|_{t=0} = p_0 v_{\text{tank}}/RT$ and $n_{\text{act}}|_{t=0} = p_{\text{atm}}Al_0/RT$, equations 8 and 9 can be solved numerically.

For the parameter values presented in Table S3, we determine the pressure-time and pressure-volume responses, calibration and sensing resolution points as shown in Fig. S12b-e. Note that for the basic interaction of the linear extension actuator with a rigid wall, assuming that the actuator comes into contact with the wall at the equilibrium state, the initial and the equilibrium internal volume of the actuator are known under different environment settings, i.e., $v_0 = Al_0$, $v_1 = A(l_0 + L)$, then we also have $dv_1/dL = A$. Therefore, we can also determine the analytical solution of the equilibrium pressure p_1 and sensing resolution dp_1/dL from equation 1 and 2 ($\xi = L$)

$$p_1 = \frac{p_0 v_{\text{tank}} + p_{\text{atm}} Al_0}{v_{\text{tank}} + A(l_0 + L)}, \quad (10)$$

$$\frac{dp_1}{dL} = -\frac{p_0 v_{\text{tank}} + p_{\text{atm}} Al_0}{(v_{\text{tank}} + A(l_0 + L))^2} A. \quad (11)$$

With the numerical simulation and analytical solution, we can qualitatively study the influence of each individual system parameter on the sensing resolution. The initial internal volume of the actuator can be changed by varying either the initial length l_0 (Fig. S12f), or the cross section A (Fig. S12g), which interestingly have different effects on

the sensing resolution. To understand this difference, we can first look at an extreme case when the volume of the actuator is infinitely large. In this extreme case, the air mass in the air tank is too small to extend the actuator, then the sensing resolution is $dp_1/dL = 0$. This vanishing of sensing resolution with an extremely large actuator can be seen from the numerical simulation results when varying either l_0 or A (dp_1/dL becomes zero after l_0 and A exceed a certain value in Fig. S12f and g), but not from the analytical solutions. This is because the analytical solution assumes that the actuator always comes into

Figure S12: **Modeling the fluidic sensing approach.** **a**, Schematics of a simplified sensing scenario: an extension actuator with a linear stiffness k comes into contact with a rigid wall from a distance L . **b-e**, Simulation results using the input values shown in Table S3: pressure-time (**b**) and pressure-volume (**c**) curves of the air tank and actuator; calibration curve of P_{eq} over L (**d**); sensing resolution curve of dP_{eq}/dL over L (**e**). **f-i** Simulation results of the effects of actuator's initial length l_0 (**f**), cross section A (**g**), stiffness k (**h**) and initial tank pressure p_0 (**i**) on the sensing resolution magnitude $|dP_{eq}/dL|$. The makers and dashed lines in **f-i** represent numerical simulation results and analytical solutions, respectively. The circular, square and triangular markers in **f-i** represent $L = 7.5, 17.5, 27.5$ mm, respectively. The results in **f-i** are based on the input values in Table S3 except that an initial tank pressure of 300 kPa (absolute pressure) is used in **g**.

contact with the wall at the equilibrium state. However, the occurrence of this contact also depends on the initial conditions. Therefore, the numerical simulation results agree with the analytical solution in Fig. S12f and g until the actuator can not reach the wall anymore at given initial conditions. Still, the analytical solution in equation 11 provides theoretical insights on how the sensing resolution is affected by l_0 and A .

Interestingly, the analytical solution also indicates that the stiffness k (Fig. S12h) does not influence the sensing resolution, because k does not affect any of the parameters in equation 11. However, when k is extremely large, the actuator becomes too stiff to extend a distance of L with given initial conditions, such that the sensing resolution becomes $dp_1/dL = 0$ as no contact occurs, which can be seen from the numerical simulation results in Fig. S12h. Therefore, a higher stiffness does require a higher tank and actuation pressure. On the other hand, the magnitude of sensing resolution increases with the initial tank pressure p_0 when the other parameters in equation 11 stay constant (Fig. S12i).

It should be noted that, in this simplified model with the linear extension actuator, we assume that the actuator volume stops changing after it hits the wall. However, in reality, the interaction with the environment does not fully constrain the deformation and thus the internal volume of the actuator, which can complicate the analysis and needs to be analyzed case by case. We believe that, despite the simplifications made in our model, it provides a framework for choosing available parameters for improving the sensing resolution.”

Response to Reviewer #2

Comment 2A

It is a solid piece of work, supported by a comprehensive amount of experimental work to demonstrate the efficacy of the fluidic sensing in soft robotics. The following comments and suggestions are offered to substantiate its contribution and significance.

1. As explicitly stated in the manuscript, the idea of the fluidic sensing is not new [34], but is re-visited in this study to “gain a better understanding of the underlying principles that allow for fluidic self-sensing, and determine if we can infer the interaction of a wide variety of soft actuator with their environment by measuring and analyzing the fluidic response of the enclosed cavity”. The honest and clear description of the literature is highly appreciated. This study takes this concept to a next level to demonstrate its validity in soft gripping/robotic applications. No previous soft robotic study investigated into the fluidic sensing concept, to the best of my knowledge.

We appreciate the Reviewer’s positive feedback about our work and the in depth comments to help us improve the manuscript. We have significantly improved our manuscript by adding new experimental and modeling results.

Comment 2B

2. How valid are the results in Figure 1, for the actuators made of different materials, especially from a flexible TPU requiring a much higher pressure to bend actuators like PeuNet, e.g., <https://doi.org/10.1089/soro.2016.0030>.

We very much appreciate the Reviewer’s interest in applying our sensing approach to 3D-printed TPU bending actuator. Following the reference provided by the Reviewer, we printed a TPU bending actuator and implemented sensing using our retrofit approach. We added the new results on the TPU bending actuator to the manuscript. With the new results, the figure about retrofitting the fluidic sensing approach became too crowded to see clearly the individual plot, therefore, we split the results into two figures (Fig. 3 and Fig. 4) in the updated manuscript. Based on these results, we added the following discussion in the main text:

“To demonstrate that the sensing approach can be retrofitted, we apply the sensing strategy to a filament actuator, a McKibben actuator, a thermoplastic polyurethane (TPU) actuator, a soft suction gripper specifically designed for medical applications and two commercially available soft grippers.”

“In all demonstrations so far, we used one or more identical soft bending actuator. However, our sensing approach can also be retrofitted to a broad range of fluidic actuators without the need for any design changes. To demonstrate the wide applicability, using our approach we sensorize a filament actuator, a McKibben actuator, a 3D-printed bending actuator (43), a suction cup, and two commercial soft grippers (Fig. 3, Fig. 4 and Supplementary Video 3).”

“To determine if our sensing approach can also be used for higher actuation pressures, we next retrofit our sensing strategy to a 3D-printed TPU bending actuator that requires

Figure 3: **Retrofitting the fluidic sensing approach to a filament actuator (a-c), a McKibben actuator (d-f) and a 3D-printed bending actuator (g-i).** The filament (a) and McKibben (d) actuator are used as a muscle to rotate an arm towards a stopper. TPU bending actuator (g) wrapping around a cylinder with a diameter D . Corresponding pressure-volume relation for the soft actuator (solid) and tank (dashed) (b, e, h) and equilibrium pressure in the system (c, f, i) for different positions of the stopper or cylinder diameter D . Experimental results from five tests are shown for each θ (b, e) and each D (h). Scale bars, 30 mm.

Figure 4: **Retrofitting the fluidic sensing approach to a suction cup (a-e) and two commercial soft grippers (f-k).** **a**, Front and bottom views of the suction gripper. Scale bar, 10 mm. **b**, Schematic of the suction gripper attaching to a soft object. **c**, Smoothed pressure-volume responses of the suction gripper (pink) and air tank (blue) when attaching to silicone samples with different shore moduli. **d**, Experimental sensing results obtained using two different initial pressures P_0 in the air tank. **e**, Force-pressure responses from three pulling tests on the suction gripper when attached to a silicone sample with a shore hardness of OO-30. Commercial vacuum (**f**) and pressurized (**i**) grippers gripping a cylindrical object with a diameter D . Scale bar, 50 mm. Corresponding pressure-volume relation for the soft gripper (solid) and tank (dashed) (**g**, **j**) and equilibrium pressure in the system (**h**, **k**) for objects. Experimental results from five tests are shown for each D (**g**, **j**).

an actuation pressure around 200 kPa (43). In previous tests with the bending actuator, we only consider a single contact between the soft actuator and the environment. Since the TPU bending actuator forms a circular shape at higher pressures (43), we tested our sensing strategy with conformal grasping (49), where the soft actuator interacts with the cylindrical object at multiple contact points (Fig. 3g). We find that the conformal grasping of cylindrical objects with various diameters results in different pressure-volume responses of the soft actuator (Fig. 3h) and that we can also correlate the equilibrium pressure with the diameter of the grasping object, even for these higher pressure ranges (Fig. 3i).”

Finally, and also based on suggestions from the other Reviewers, we also added a model of the fluidic sensing approach based on a simplified interaction of a linear extension actuator with the environment to give a better understanding how various parameters affect the sensing resolution (see the response to Comment 1B). Following these results, we found that in principle our sensing approach and the sensing resolution do not depend on stiffness (although we will have to use higher pressures to inflate the actuators, and different external sensors with larger pressure range).

Comment 2C

3. Please discuss how the sensing/calibration curves change for the actuators requiring relatively higher input pressures? As stated for the filament actuator, less sensing resolution will be achieved for these actuators.

As pointed out by the Reviewer, we observe that for the filament actuator less sensing resolution is achieved compared to the McKibben actuator. Even though the two types of actuators were tested on the same artificial muscle setup, there are many differences between the two actuators, including the internal volume and actuation pressure of the actuator, which make it difficult to evaluate the influence of each parameter and pinpoint the main reason for the difference between the sensing resolution of both actuators.

Therefore, to study the effect of parameters on the sensing resolution, we included a model of the fluidic sensing approach based on a simplified interaction of a linear extension actuator with the environment in the article, as discussed in Comment 1B. Our main findings from this model are that stiffness does not directly influence the sensing resolution, but the design of the actuator does considerably influence the sensing resolution.

Comment 2D

4. While it is appreciated to apply this concept to different actuators, there is no indication of the actuators typically requiring higher pressures ($> 250\text{kPa}$). These actuators will typically show smaller differences in their pressure-volume curves. Yes, the filament actuator requires pressures at this level. However, the volume of these actuators is much smaller to result in enough change in the pressure-volume curves. My point is more for actuators with a larger volume and requiring a high pressure in the range of $> 250\text{kPa}$.

We thank the Reviewer for pointing out the actuator design parameters that may influence the sensing performance. We added to the manuscript the test results of a 3D-printed TPU actuator ($\sim 200\text{ kPa}$) that has a larger volume, as shown in the response to Comment 2B. For the influence of actuator volume and actuation pressure on the sensing resolution, we refer to our response to Comment 1B. It is important to note that even though the relative pressure difference (pressure

difference due to interaction with the environment in comparison to maximum pressure obtained in the actuator) might be smaller for stiffer actuators, the absolute pressure change for all actuators we tested is in the same order of magnitude (Table 1). Luckily, for these absolute pressure differences there are ample (relatively cheap) pressure sensors on the market that span this full range of pressures with high accuracy (Table S2), so that the sensing resolution is not affected by a lower relative pressure difference.

Comment 2E

5. *Although the effect of the internal volume of the air tank on the tank's pressure-volume response is being discussed or remarked, how about the internal volume of the PneuNet like actuators? Also, how about PneuNet like actuators with a 3D movement such as bending and twisting motions simultaneously, (e.g., <https://doi.org/10.1089/soro.2019.0015>) as opposed to 2D bending movement of PneuNet actuators?*

We hope that the model implemented in Comment 1B and the previous answers to the questions from Reviewer 2 give sufficient insight into the effect of actuator volume. According to equation 2 in the response to Comment 1B, the sensing resolution is influenced by the final internal volume v_1 of the actuator after the interaction with the environment and its derivative over the sensing target $dv_1/d\xi$. Yet, there is no trivial relationship between the initial volume of the actuator and the sensing resolution, as, e.g., the length and area affect sensing resolution differently (Fig. S12f and g) and the response of $dv_1/d\xi$ is often non-linear. For example, for the modeled linear extension actuator in the response to Comment 1B, an increase in length (increase in initial volume) reduces the sensing resolution, while an increase in area (also an increase in initial volume) first increases and then decreases the sensing resolution.

Following the results above, it is not straightforward to determine the influence of 3D movement on the sensing resolution, as it depends on how the 3D movement affects v_1 and $dv_1/d\xi$. Even though we demonstrate that we can retrofit sensing to pneumatic actuators, optimizing the sensing resolution by for example changing the tank size and initial tank pressure should be done on a case by case basis.

We added the citation and following discussion to the conclusion of the manuscript:

“According to equation 2, the sensing resolution is influenced by the final internal volume v_1 of the actuator after the interaction with the environment and its derivative over the sensing target $dv_1/d\xi$. Yet, there is no trivial relationship between the initial volume of the actuator and the sensing resolution. Even though we demonstrate that we can retrofit sensing to pneumatic actuators, optimizing the sensing resolution by for example changing the tank size and the initial tank pressure should be done on a case by case basis, and can best be done by experimentally obtaining the relation between pressure and volume for specific interactions with the environment. For example, if we want to apply our strategy to pneumatic actuators that can generate complex motions with multiple degrees of freedom (56), the sensing resolution depends on how the interaction of the actuator with the environment affects v_1 and $dv_1/d\xi$ in equation 2, which is not straightforward to predict beforehand and depends on the application.”

Comment 2F

6. *When picking up a tomato or similar with more softness (especially softer than the gripper), how will the closed-loop control approach will work? Reaching a larger threshold may damage the produce softer object or agricultural produce.*

This is an interesting question brought forward by the Reviewer, and also relates to a point raised in comment 3G. To highlight the potential for working with objects of different stiffness, we added a third closed-loop control demonstration for tomato sorting, where we identify and remove over-ripe tomato with our fluidic sensing approach. We demonstrate that our approach can consistently identify the overripe tomato.

We added Fig. 6 and the following discussions to the manuscript. Moreover, we also added Supplementary Video 6 to show the tomato ripeness sensing experiments. We also attached the Python code and output from Jupyter Notebook in Appendix A of this response letter to provide the Reviewers with more details about the tomato ripeness sensing experiment. We will upload all the raw data and codes used in this work to Zenodo, an open repository, before publication.

“In the third closed-loop control experiment (Fig. 6 and Supplementary Video 6), we detect the ripeness of tomatoes by applying the method mentioned in Fig. 2f and g to estimate the indentation depth. We demonstrate that the proposed sensing approach can provide feedback for automated sorting of an overripe tomato from ripe tomatoes. For versatility, we select the commercial vacuum gripper for this demonstration, also as it has the largest sensing resolution (Table 1). The demonstration includes one cycle of calibration and five separate cycles of sensing to determine repeatability. The positions of the four tomatoes (including one overripe tomato) and one dummy are shuffled randomly between sensing cycles. All size predictions are based on one calibration process with 3D-printed rigid dummies, where the gripper pressure P_{gripper} is compared to a reference response P_{ref} to determine $\Delta P = P_{\text{gripper}} - P_{\text{ref}}$ over time for different diameters of the object that is being gripped (Fig. 6b and c). As explained by the stiffness sensing method in Fig. 2f and g, the size of the tomato upon gripping can be inferred by the time of first contact t_c (Fig. 6d), and the size at equilibrium can be inferred by the ΔP at equilibrium (Fig. 6e).

It is important to note that the method that uses the time of first contact is strongly affected by the alignment of the tomato inside the gripper, leading to an early rise of ΔP -time curve and inaccurate size predictions of the tomato upon gripping and sensing success rates of 40% and 45% (Fig. 6f and g) for the rigid dummy and tomatoes, respectively. To increase the sensing success rate, we can perform the gripping event twice, the first to center the object inside the gripper, and the second gripping event to extract sensing feedback (Fig. 6h and i). Alternatively, we can choose t_c at higher ΔP values for both calibration and sensing (Fig. 6j and k), so that the object has been effectively centered during sensing. With t_c at $\Delta P = 2\text{kPa}$, sensing during either the first or second gripping event gives 100% success rates for both rigid dummy and tomatoes (Fig. 6j-m). We can also infer the initial size of the tomato by ΔP_{max} (Fig. S19 and Supplementary Video 6) instead of t_c to avoid the influence of misalignment, which gives 100% and 90% success rate for rigid dummy and tomatoes, respectively. While we were able to pick out the rotten tomato consistently, since the gripper squeezes tomatoes for ripeness detection in this method, post-harvest studies should be performed in the future to avoid extra damage to the produce when applying the method in practical applications.”

Figure 6: **Picking out an override tomato with fluidic sensing.** **a**, Snapshots of the closed-loop control demonstration. Five 3D-printed rigid dummies ($D = 45, 50, 55, 60, 65$ mm) are used for calibration. The other five objects are tested for sensing, including three ripe tomatoes, one override tomatoes and one dummy with a diameter $D = 50$ mm. The whole demonstration includes one cycle of calibration and five cycles of sensing. In each sensing cycle, two gripping events are carried out for each object. The positions of the objects are shuffled randomly between cycles. **b-e**, Calibration results. Experimental results from three measurements are plotted for each D in **b,c**. The error bars in **d,e** represent the standard deviation of three measurements. **f-m**, Sensing results. The solid line in **f,h,j,l** represents that the object diameter inferred by t_c equals that inferred by ΔP_{eq} , the dashed line represents the object diameter inferred by ΔP_{eq} is 3 mm smaller than that inferred by t_c . For the ripe tomatoes and rigid dummy, a measurement above the dashed line is considered as a successful sensing event. Conversely, for the override tomato, a measurement below the dashed line indicates sensing success.

Figure S19: **Picking out the overripe tomato with fluidic sensing based on calibrations with ΔP_{\max} and ΔP_{eq} .** **a**, Snapshots of the closed-loop control demonstration (Demo 2 in Supplementary Video 6). **b-d**, Calibration results. Experimental results from three measurements are plotted for each D in **b**. The error bars in **c,d** represent the standard deviation of three measurements. **f-m**, Sensing results. The solid line in **e** represents that the object diameter inferred by ΔP_{\max} equals that inferred by ΔP_{eq} , the dashed line represents the object diameter inferred by ΔP_{eq} is 0.5 mm smaller than that inferred by ΔP_{\max} .

Response to Reviewer #3

Comment 3A

This work introduces an innovative self-sensing strategy for soft fluidic actuators, utilizing the measurement of internal pressure variations within actuator cavities. These pressure changes are responsive to interactions with the environment, such as contact with surfaces of different textures or objects with varying dimensions. The key advantage of this approach lies in its simplicity, as it eliminates the need for sensor redesign or compromises in compliance due to the integration of rigid sensing components. The presented method demonstrates versatility by showcasing its applicability to a wide range of fluidic actuators, including commercially available ones. The paper incorporates precision experiments with meticulously controlled parameters, covering diverse actuation and sensing protocols. For instance, it investigates scenarios where a soft actuator makes contact with the ground at different heights or interacts with regular geometrical objects (e.g., blocks, cylinders) of varying dimensions. By comparing the pressure difference changes to a reference configuration (freely moving actuator), the authors utilize this feedback to derive tactile information. The work demonstrates comprehensiveness, with precise experimental execution and compelling demonstrations that substantiate the proposed concept.

We thank the Reviewer for the positive feedback and recognizing the simplicity and elimination of sensor redesign as the key advantage of our approach. We really appreciate the in depth comments from the Reviewer which helped us further substantiate our retrofit sensing concept.

Comment 3B

However, there are certain considerations to be addressed before the publication of this work in Nature Communications. The foremost concern is the relatively low range of pressure change observed during robot-environment interactions ($\sim 1\text{kPa}$). This necessitates the use of highly accurate pressure sensors and may overlap with naturally occurring pressure variations in real-world scenarios.

We thank the Reviewer for pointing out the concern of relatively low range of pressure change during robot-environment interactions. Even though the absolute pressure variations are relatively low, we did not have to use expensive and very precise pressure sensors, as hopefully clarified by our answer to this comment and the additions made to the manuscript.

First, to give better insight into the pressure variations in the different actuators, we added Table 1 (see the answers in Comment 1B) to provide an overview of the sensing performance of various actuators and grippers tested in this work. The absolute pressure change observed during robot-environment interactions ranges from 1.3 kPa to 5.9 kPa among the soft actuators and grippers we tested. Even though the relative pressure difference (pressure difference due to interaction with the environment in comparison to maximum pressure obtained in the actuator) might be smaller for actuators that require higher inflation pressure (e.g., TPU and filament actuators), the absolute pressure change for all actuators we tested is in the same order of magnitude. Note that the average sensing resolution can be determined by dividing the absolute pressure change by the tested range of sensing target and is therefore not affected by a lower relative pressure difference. We also find that the sensing resolution is determined by the total amount of air in the system, the internal geometric volume of the tank and soft actuator at equilibrium, and the sensitivity of the internal geometric

volume of the actuator to the sensing target. We added a new section **Characterizing the sensing resolution** to the manuscript. We refer to our answers in Comment 1B.

The overall sensing accuracy, on the other hand, is determined by both the sensing resolution of the actuator or gripper and the sensing accuracy of the pressure sensor used, and luckily, there are ample (relatively cheap) pressure sensors on the market that span various pressure ranges with high enough accuracy for our purpose (Table S2, see the answers in Comment 1B). For example, in the tests with the PneuNet actuator in Fig. 1b, we used a $\pm 34.5\text{kPa}$ pressure sensor with an accuracy of $\pm 0.25\%$ (of Full Scale Span). By dividing the pressure sensor error ($\pm 0.1725\text{kPa}$) by the average sensing resolution (0.04kPa/mm), we can obtain an overall sensing accuracy of $\pm 4.31\text{mm}$ for the PneuNet actuator. This sensing accuracy is valid for one-time measurement with the pressure sensor. In this work, however, we always do the pressure measurement over a period of time (which we were able to reduce to 0.05 seconds in Fig. S20) and calculate the average value, which gives a higher overall sensing accuracy, e.g., $\pm 1.7\text{mm}$ with a 95% confidence interval (Fig. S3a) for the PneuNet actuator.

Finally, it should be pointed out that all the tests in this work were done in standard laboratory environment. Even in this controlled environment we observed that our results were affected by the output pressure variations coming from the pressure regulator (Fig. S7). Naturally occurring pressure variations should definitely be considered when applying our sensing strategy in real-world environments. For example, in a farm, the temperature change during a day can cause pressure variations in the soft gripper. While this should be investigated further, we do believe that since the temperature change happens at a time scale much slower than the gripping event, we can first perform an empty gripping event as a reference and then compare the sensing response to the reference response, as described in Fig. S7 in the original manuscript, to reduce the influence of temperature-induced pressure variations. Other environmental variations could be taken into account through similar considerations.

Comment 3C

To improve the paper, I suggest addressing the following comments:

1. *The authors have based their fluidic sensing on the ideal gas law, but it would be beneficial to discuss whether similar pressure fluctuations could occur in liquid-driven actuators, where the fluid is incompressible (e.g., with water). Clarifying the applicability of this method to liquid-driven actuators would enhance the paper's completeness.*

We thank the Reviewer for pointing out the potential applicability of the sensing strategy to liquid-driven actuators. The tank method (Fig. 1b) used mostly in this manuscript is not directly applicable for the sensing, as it depends on the compressibility of air and the final balance achieved by the volume in the tank and the volume in the actuator. This could be solved by using a flexible tank (e.g., a balloon), such that the compressibility of the air is replaced by the elasticity of the tank. A simpler approach could instead be to use the pressure control (Fig. 1c) or flow control (Fig. 1d) method to obtain sensory feedback from the liquid-driven actuator's interaction with the environment. We added the discussion to the conclusion of the manuscript:

“While we were able to retrofit our sensing approach to a range of soft actuators and grippers, it should be noted that the air tank method in Fig. 1b is not directly applicable to soft actuators driven by incompressible liquid, as the method depends on the compressibility of air and the final pressure balance between the tank and actuator. This could be solved by using a flexible tank (e.g., a balloon), such that the compressibility of the air is replaced by the elasticity of the tank. A simpler approach could instead be

to use the pressure control (Fig. 1c) or flow control (Fig. 1d) method to obtain sensory feedback from the liquid-driven actuator’s interaction with the environment.”

Comment 3D

2. While various actuators were examined in this work, the pressure difference observed in most cases was approximately 1 kPa. It would be insightful to investigate the effect of actuator stiffness on the sensitivity of the proposed technique. This analysis would provide valuable insights into the relationship between actuator properties and the sensing strategy’s effectiveness.

We really appreciate the Reviewer’s question about the influence of actuator design parameters on the sensing performance, which was also pointed out by the other Reviewers. Based on this and their comments, we added a model of the fluidic sensing approach based on a simplified interaction of a linear extension actuator with the environment, and gave qualitative analysis of the influence of various parameters, such as actuator volume, stiffness and actuation pressure, on the sensing resolution. Moreover, we performed additional experiments on a stiffer TPU bending actuator. We therefore refer to our answers in Comment 1B and in Comment 2B and 2D.

Comment 3E

3. Although the experiments conducted in this study were primarily under quasistatic conditions, it is important to consider real-world applications where actuation occurs at higher speeds (e.g., Soft Robotics Inc. pick-and-place demonstrations). It would be valuable to evaluate the performance of the sensing strategy at different actuation rates, particularly for simple cases. This analysis would further demonstrate the robustness of the proposed approach.

The sensing speed is indeed an important aspect to address to give insights into the sensing robustness. We performed additional experiments with higher actuation rates, and added these new results to the manuscript to show the influence of actuation rate on the sensing performance of the soft gripper. In these experiments, we removed the flow resistor between the tank and the soft gripper (Fig. S20a-b) to achieve fast actuation. While, as expected, the pressure response for slow and fast actuation is highly affected by this resistance (Fig. S20c-e), the difference in pressure response for different object is sufficient to predict the diameter of the object that is being sensed (Fig. S20f). Focusing on the fast actuation, we can vary the waiting time after opening to valve to obtain a pressure measurement. We find that, even though the pressure varies over time, the influence of the object diameter on the pressure is sufficient to already distinguish the objects 0.5 seconds after initiating the actuation (opening the valve), as shown in Fig. S20g-h.

Furthermore, we implemented fast sensing in the sorting demo to determine the robustness, where the sensing feedback is collected 0.5s (instead of 5s) after the opening of the valve between the air tank and actuator.

The following results and texts are added to the main manuscript:

“It should be noted that all the closed-loop control demonstrations above were performed under quasistatic conditions. We tested the sensing strategy at different actuation speeds (Fig. S20) and find that the effectiveness of the sensing strategy is not affected by the actuation speed, as long as the sensing feedback is collected after the actuator comes into contact with the environment. To speed up the sensing process for real-world applications, it is important to ensure an actuation speed that is high enough for the interaction

Figure S20: **Influence of the actuation rate on the D -pressure calibrations of the soft gripper.** **a, b,** We vary the actuation rate by placing a flow resistor ($R = 5.7 \times 10^8 \text{ Pa} \cdot \text{s}/\text{m}^3$) between the solenoid valve and the gripper. **c-e,** Experimental results of the soft gripper gripping cylindrical objects with a diameter $D = 40$ mm, 60 mm, 80 mm and 100 mm. The dashed and solid curves represent the tests in slow and fast actuation configurations, respectively. A total of eight actuation cycles were performed on each cylindrical object, and the last three cycles are shown in **c** and **d**. **f,** D -pressure calibrations in slow and fast actuation configurations. Each data point represents the average value of the equilibrium pressure in the last five actuation cycles. For each actuation cycle, the equilibrium pressure is averaged over a 5 s period starting at 10 s after the actuation. **g, h,** Pressure measurements in fast actuation configuration and D -pressure calibrations with different waiting time after the actuation. Results from the last five actuation cycles are superimposed in **g** for each gripping object. Each data point in **h** represents the average value of the gripper pressure in the last five actuation cycles. For each actuation cycle, the pressure in the gripper is averaged over a 0.05 s period starting at 0.2 s, 0.3 s, 0.5 s, 2 s, 5 s after the actuation (the opening of the valve in **b**). The error bar represents the standard deviation of the gripper pressure in the last five actuation cycles.

with the environment to happen before the collection of fluidic sensing feedback. To prove the feasibility, we successfully implemented fast sensing in the size sorting demonstration, where pressure values are collected 0.5 s (instead of 5 s) after the opening of the valve between the air tank and gripper (Fig. S21 and Supplementary Video 4).”

Figure S21: **Sorting experiments with a faster speed.** The input order of cylindrical objects is 60 mm, 100 mm, 80 mm, 40 mm in **a** and 100 mm, 40 mm, 60 mm, 80 mm in **b**. For each measurement event, the equilibrium pressure was averaged over a 0.05 s period starting at 0.5 s after the actuation, as indicated by the red bands in the plot. The four pressure feedback measurements (average \pm standard deviation) in **a** are 38.21 ± 0.11 kPa, 38.84 ± 0.07 kPa, 38.59 ± 0.07 kPa, 37.44 ± 0.07 kPa, respectively. The four pressure feedback measurements (average \pm standard deviation) in **b** are 39.17 ± 0.11 kPa, 37.33 ± 0.06 kPa, 37.92 ± 0.07 kPa, 38.52 ± 0.06 kPa, respectively. Note that the initial tank pressure was set at 74 kPa here to ensure successful gripping of each object, especially the one with a diameter of 40 mm. Note that the coordinates of the robotic arm were intentionally not saved in these experiments in order to obtain a high data frequency (~ 87 Hz) that allows the fast sensing over a 0.05s period.

Comment 3F

4. *In the case of commercial actuators, it is notable that the effect of pressure difference is more pronounced in vacuum-driven soft grippers compared to positive pressure actuators (Fig. 3m vs Fig. 3p). It would be beneficial to provide insights into this discrepancy and explore the underlying factors contributing to the varying response between these two types of actuators.*

We appreciate the Reviewer’s interest in the discrepancy between the two commercial grippers. Fig. 3m and p in the original manuscript provided a general picture of how the variations in pressure-volume curves between the two commercial grippers lead to different sets of equilibrium points based on the conservation of air mass. Since there are also many other factors involved when comparing the two commercial grippers, such as the internal geometric volume of the air tank and gripper, we added new results and discussions based on the ideal gas law to explore the influence of these factors on the varying response between the two commercial grippers. We further refer to the response to Comment 1B.

Comment 3G

5. *The closed-loop control in this work is based on a simple pressure difference thresholding. Considering the potential of the proposed self-sensing technique, it would be interesting to investigate whether richer information beyond pressure change can be extracted. Expanding on the capabilities of the sensing strategy to gather more comprehensive data would add depth to the study.*

6. *The tomato picking demonstration showcased in the paper is intriguing. Given that the actuator can discern surfaces with different stiffness (e.g., Ecoflex vs. rigid surfaces), it would be compelling to explore the application of this method in sorting ripened from unripened tomatoes. This extension of the work would demonstrate its practicality in agricultural automation and highlight its potential for addressing real-world challenges.*

We appreciate the Reviewer’s interest in our closed-loop control demonstrations. We added a third closed-loop control demonstration of picking out overripe tomato with fluidic sensing. In this demonstration, both time of contact and pressure change are extracted to sense the ripeness of tomatoes, we hope that this additional experiment gives more insight into the practical and broader applicability of our approach. We further refer to our response to Comment 2F.

Comment 3H

7. *If applicable, please report the success rate of the tomato picking experiment. Including this information would provide valuable insights into the performance of the sensing strategy in a specific application scenario.*

We appreciate the Reviewer’s interest in the success rate of the tomato picking experiment. We added this information and a discussion to the manuscript:

“We tested a total of nine tomatoes in three runs, out of which six tomatoes were successfully picked and placed, one tomato was not picked by the gripper, the other two were successfully picked but not recognized because the tomato slipped into the palm of the gripper after being picked from the stem, resulting in a ΔP smaller than the

Figure S18: **Snapshots of the tomato picking experiments and gripper pressure and TCP coordinates over time during the picking experiments (Demo 2 and 3 in Supplementary Video 5).** The red band represents pressure feedback measurement, and the yellow band represents tomato placement. During the picking of the third tomato in **a** and the second tomato in **b**, the tomato slipped into the palm of the soft gripper, resulting in $\Delta P = 0.1$ kPa which is smaller than the threshold (0.2 kPa). The tomato picking was not recognized and the gripper dropped the tomato and started a new picking attempt. The gripper failed to pick the third tomato in **b**.

threshold (Fig. S18 and Supplementary Video 5). As our soft gripper was not specifically designed for picking tomatoes, the design of the gripper should be optimized for this task. Importantly, this would not affect the retrofit implementation of our sensing approach.”

To address the Reviewer’s concern about sensing success rate, we also reported the success rate of the newly-added tomato ripeness sensing demonstration, see our response to Comment 2F.

Comment 3I

By addressing these comments and further expanding the paper's scope, this work has the potential to make a significant contribution to the field of self-sensing in fluidic actuators.

We would like thank the Reviewer again for the in-depth comments that helped us improve the manuscript significantly. We hope the Reviewer agrees that with the additional experiments and model, we have adequately addressed the concerns raised by the Reviewer.

Appendix A

The data in this appendix is to provide the Reviewers with more details about the tomato ripeness sensing experiment. We will upload all the raw data and codes used in this work to the open repository Zenodo before publication.

Supplemental data for the tomato ripeness sensing experiment

Python code

```
robot = TryConnectionUntilSuccess()
robot.GoSoftHome()
eqtime = 20 # gripper closing time
relaxtime = 20 # gripping opening time
TC_threshold = 2 # extract t_c at delta_p = 2 kPa
robot.StartDataLogger(save=True)
v=1 # robotic arm velocity
a=0.5 # robotic arm acceleration
Dia = [45, 50, 55, 60, 65] # calibration dummy diameter
[DP_AVG,DP_STD,TC_AVG,TC_STD,DPMAX_AVG,DPMAX_STD] = Softnesscalibration(
inputorder = Dia,
eqtime=eqtime,
relaxtime=relaxtime,
TC_threshold = TC_threshold)
plt.figure(figsize=[14,6])
plt.title("GrabAndMeasureoverTime",fontsize=20)
for n in range(5):
robot.GoSoftHome(v=v,a=a)
robot.MoveRelative(dx=.12,v=v,a=a)
D_DP=[]
D_TC=[]
D_MAX=[]
for i in range(5):
squareside = .115
dz_disp_dist = .156
if i > 0:
dy_disp = squareside
else:
dy_disp = 0
robot.MoveRelative(dy=dy_disp,v=v,a=a)
# reference measurement, gripper closes in the air
[t_ref, p_ref] = robot.GrabAndMeasureoverTime(measuretime=eqtime)
robot.OpenGripper()
time.sleep(relaxtime)
robot.MoveRelative(dz=-dz_disp_dist,v=v,a=a)
# first gripping event
robot.CloseGripper()
```

```

time.sleep(eqtime)
robot.OpenGripper()
time.sleep(relaxtime)
DP = []
TC = []
DPMAX = []
# second gripping event
for _ in range(1):
[t_tomato, p_tomato] = robot.GrabAndMeasureoverTime(
measuretime=eqtime)
delta_p = p_tomato - np.interp(t_tomato, t_ref, p_ref)
plt.plot(t_tomato, delta_p, 'o-')
# extract delta_p at equilibrium
delta_p_DP = delta_p[t_tomato>eqtime-1]
delta_p_DP = delta_p_DP[t_tomato<eqtime-0.5]
DP.append(np.mean(delta_p_DP))
# extract t_c at delta_p = TC_threshold
delta_p_TC = delta_p[t_tomato>1]
delta_p_TC = delta_p_TC[t_tomato<7]
t_tomato_TC = t_tomato[delta_p_TC[
delta_p_TC>TC_threshold].index[0]]
TC.append(t_tomato_TC)
# extract the peak value of delta_p
delta_p_MAX = delta_p[t_tomato>1]
delta_p_MAX = delta_p_MAX[t_tomato<15]
DPMAX.append(np.max(delta_p_MAX))
D_DP.append(np.interp(np.mean(DP), DP_AVG, Dia))
print("Tomato No.: ", i+1)
print("Delta P:", np.mean(DP),
", std: ", np.std(DP),
"; Inferred diameter: ", D_DP[i])
D_TC.append(np.interp(np.mean(TC), TC_AVG[:-1], Dia[:-1]))
print("Time of contact:", np.mean(TC),
", std: ", np.std(TC),
"; Inferred diameter: ", D_TC[i])
Delta_D = np.interp(np.mean(TC), TC_AVG[:-1], Dia[:-1])
- np.interp(np.mean(DP), DP_AVG, Dia)
print("D_TC - D_DP = ", Delta_D, ' mm')
D_MAX.append(np.interp(np.mean(DPMAX), DPMAX_AVG, Dia))
print("Delta P MAX: ", np.mean(DPMAX),
", std: ", np.std(DPMAX),
"Inferred diameter: ", D_MAX[i])
Delta_D_max = np.interp(np.mean(DPMAX), DPMAX_AVG, Dia)
- np.interp(np.mean(DP), DP_AVG, Dia)
print("D_MAX - D_DP = ", Delta_D_max, ' mm')
if Delta_D >3: # demo 1 in Supplementary Video 6
#if Delta_D_max > 0.5: # demo 2 in Supplementary Video 6
robot.MoveRelative(dz=dz_disp_dist,v=v,a=a)
robot.MoveRelative(dx=.12,v=v,a=a)
robot.MoveRelative(dz=-dz_disp_dist,v=v,a=a)
robot.OpenGripper()
time.sleep(relaxtime)

```

```

robot.MoveRelative(dz=dz_disp_dist,v=v,a=a)
robot.MoveRelative(dx=-.12,v=v,a=a)
else:
robot.OpenGripper()
time.sleep(relaxtime)
robot.MoveRelative(dz=dz_disp_dist,v=v,a=a)
robot.GoSoftHome(v=v,a=a)
plt.grid()
plt.ylabel("Delta P (kPa)",fontsize=18)
plt.xlabel("Time (s)",fontsize=18)
robot.EndDataLogger()
def Softnesscalibration(inputorder=[45,50,55,60,65],
eqtime = 20,
relaxtime = 20,
TC_threshold = 2,
a=0.5,
v=1):
robot.GoSoftHome(v=v,a=a)
fig1, ax1 = plt.subplots(figsize=(14, 6))
plt.ylabel("Delta P (kPa)",fontsize=18)
plt.xlabel("Diameter (mm)",fontsize=18)
fig2, ax2 = plt.subplots(figsize=(14, 6))
plt.ylabel("Time of contact (s)",fontsize=18)
plt.xlabel("Diameter (mm)",fontsize=18)
fig3, ax3 = plt.subplots(figsize=(14, 6))
plt.ylabel("Delta P max (kPa)",fontsize=18)
plt.xlabel("Diameter (mm)",fontsize=18)
# dummy
robot.CloseGripper()
time.sleep(eqtime)
robot.OpenGripper()
time.sleep(relaxtime)
# reference measurement, gripper closes in the air
[t_ref, p_ref] = robot.GrabAndMeasureoverTime(measuretime=eqtime)
robot.OpenGripper()
time.sleep(relaxtime)
# calibration begins
DP_AVG = []
DP_STD = []
TC_AVG = []
TC_STD = []
DPMAX_AVG = []
DPMAX_STD = []
for obj in range(5):
print("\n# obj", obj+1)
squareside = .115
dz_disp_dist = .156
dy_disp = squareside
if obj > 0:
dy_disp = squareside
else:
dy_disp = 0
robot.MoveRelative(dy=dy_disp,v=v,a=a)

```

```

robot.MoveRelative(dz=-dz_disp_dist,v=v,a=a)
DP = []
TC = []
DPMAX = []
# first gripping event to center the dummy
robot.CloseGripper()
time.sleep(eqtime)
robot.OpenGripper()
time.sleep(relaxtime)
# three gripping events to collect data for calibration
for _ in range(3):
[t_meas, p_meas] = robot.GrabAndMeasureoverTime(
measuretime=eqtime)
robot.OpenGripper()
time.sleep(relaxtime)
# extract delta_p at equilibrium
delta_p = p_meas - np.interp(t_meas, t_ref, p_ref)
delta_p_DP = delta_p[t_meas>eqtime-1]
delta_p_DP = delta_p_DP[t_meas<eqtime-0.5]
DP.append(np.mean(delta_p_DP))
# extract t_c at delta_p = TC_threshold
delta_p_TC = delta_p[t_meas>1]
delta_p_TC = delta_p_TC[t_meas<7]
t_meas_TC=t_meas[delta_p_TC[delta_p_TC>TC_threshold].index[0]]
TC.append(np.mean(t_meas_TC))
# extract the peak value of delta_p
delta_p_MAX = delta_p[t_meas>1]
delta_p_MAX = delta_p_MAX[t_meas<15]
DPMAX.append(np.max(delta_p_MAX))
DP_AVG.append(np.mean(DP))
DP_STD.append(np.std(DP))
TC_AVG.append(np.mean(TC))
TC_STD.append(np.std(TC))
DPMAX_AVG.append(np.mean(DPMAX))
DPMAX_STD.append(np.std(DPMAX))
print("Object diameter: ", inputorder[obj], ' mm')
print("Delta p: ", np.mean(DP), "std: ", np.std(DP))
print("Time of contact: ", np.mean(TC), "std: ", np.std(TC))
print("Delta p max: ", np.mean(DPMAX),"std: ", np.std(DPMAX))
del DP,TC,DPMAX
robot.MoveRelative(dz=dz_disp_dist,v=v,a=a)
robot.GoSoftHome(v=v,a=a)
D=inputorder
ax1.errorbar(D, DP_AVG, DP_STD, DP_STD, 'ko')
ax2.errorbar(D, TC_AVG, TC_STD, TC_STD, 'rs')
ax3.errorbar(D, DPMAX_AVG, DPMAX_STD, DPMAX_STD, 'bo')
plt.show()
return DP_AVG, DP_STD, TC_AVG, TC_STD, DPMAX_AVG, DPMAX_STD

```

Output from Jupyter Notebook (Demo 1 in Supplementary Video 6)

Connected after 1 attempts
Gripper already open

```
# obj 1
Object diameter: 45 mm
Delta p: 1.8119906423412309 std: 0.004301896515751774
Time of contact: 5.8899412000003695 std: 0.02172117183686668
Delta p max: 4.730344839897498 std: 0.036776013100570146
```

```
# obj 2
Object diameter: 50 mm
Delta p: 2.2239130592089875 std: 0.0035108804320289153
Time of contact: 5.563874666666986 std: 0.010324471409773066
Delta p max: 5.442913455914145 std: 0.016402639608483448
```

```
# obj 3
Object diameter: 55 mm
Delta p: 2.628139293232808 std: 0.002858208723654545
Time of contact: 5.214546833332861 std: 0.0005608616128303366
Delta p max: 6.125743693687723 std: 0.00780235643265694
```

```
# obj 4
Object diameter: 60 mm
Delta p: 3.047912466707889 std: 0.004590091249658994
Time of contact: 4.894911533333773 std: 0.00340529559368619
Delta p max: 6.80802318666979 std: 0.016158505483154284
```

```
# obj 5
Object diameter: 65 mm
Delta p: 3.456193798413015 std: 0.0017019582619798317
Time of contact: 4.561784733332995 std: 0.011464532352529798
Delta p max: 7.473484906284177 std: 0.016261437788063098
```

Tomato No.: 1
Delta P: 2.7386726844363745 , std: 0.0 ; Inferred diameter: 56.316584743714316
Time of contact: 5.094889099998909 , std: 0.0 ; Inferred diameter: 56.87178533369583
D_TC - D_DP = 0.5552005899815171 mm
Delta P MAX: 6.368439106033289 , std: 0.0 Inferred diameter: 56.778562999781855
D_MAX - D_DP = 0.46197825606753895 mm

Tomato No.: 2
Delta P: 2.533444229589802 , std: 0.0 ; Inferred diameter: 53.82868730833752
Time of contact: 5.011234400000831 , std: 0.0 ; Inferred diameter: 58.180381411762255
D_TC - D_DP = 4.351694103424734 mm
Delta P MAX: 6.131017244599203 , std: 0.0 Inferred diameter: 55.038646558820275
D_MAX - D_DP = 1.209959250482754 mm

Tomato No.: 3
Delta P: 2.2227115094473824 , std: 0.0 ; Inferred diameter: 49.98541533900071
Time of contact: 5.572200599999633 , std: 0.0 ; Inferred diameter: 49.872327692640965
D_TC - D_DP = -0.11308764635974455 mm
Delta P MAX: 5.38680094605936 , std: 0.0 Inferred diameter: 49.606265918863635
D_MAX - D_DP = -0.3791494201370753 mm

Tomato No.: 4
Delta P: 2.5970790025769426 , std: 0.0 ; Inferred diameter: 54.61580560535768
Time of contact: 5.11052699999971 , std: 0.0 ; Inferred diameter: 56.626762334028605
D_TC - D_DP = 2.0109567286709265 mm
Delta P MAX: 6.173568049191136 , std: 0.0 Inferred diameter: 55.35047481270751
D_MAX - D_DP = 0.7346692073498318 mm

Tomato No.: 5
Delta P: 3.170828113919178 , std: 0.0 ; Inferred diameter: 61.50528125665151
Time of contact: 4.7429424000001745 , std: 0.0 ; Inferred diameter: 62.28095027679015
D_TC - D_DP = 0.7756690201386434 mm

Delta P MAX: 7.054394598306246 , std: 0.0 Inferred diameter: 61.851131360187175
D_MAX - D_DP = 0.3458501035356676 mm
Tomato No.: 1
Delta P: 2.728545841649399 , std: 0.0 ; Inferred diameter: 56.19596194756061
Time of contact: 5.092892900000152 , std: 0.0 ; Inferred diameter: 56.90301154680125
D_TC - D_DP = 0.7070495992406407 mm
Delta P MAX: 6.326266821389606 , std: 0.0 Inferred diameter: 56.469508682031815
D_MAX - D_DP = 0.2735467344712035 mm
Tomato No.: 2
Delta P: 3.1930593409229138 , std: 0.0 ; Inferred diameter: 61.77753503459049
Time of contact: 4.728725299999496 , std: 0.0 ; Inferred diameter: 62.49433899244805
D_TC - D_DP = 0.7168039578575574 mm
Delta P MAX: 7.045811744124006 , std: 0.0 Inferred diameter: 61.786643396948556
D_MAX - D_DP = 0.009108362358063005 mm
Tomato No.: 3
Delta P: 2.4887330309452076 , std: 0.0 ; Inferred diameter: 53.27564058745151
Time of contact: 5.067436699999234 , std: 0.0 ; Inferred diameter: 57.30121850330746
D_TC - D_DP = 4.025577915855948 mm
Delta P MAX: 6.0937879787966835 , std: 0.0 Inferred diameter: 54.76600542035723
D_MAX - D_DP = 1.490364832905719 mm
Tomato No.: 4
Delta P: 2.6379555642701074 , std: 0.0 ; Inferred diameter: 55.11692351557433
Time of contact: 5.186305299999731 , std: 0.0 ; Inferred diameter: 55.44177744656505
D_TC - D_DP = 0.32485393099072013 mm
Delta P MAX: 6.079923025019969 , std: 0.0 Inferred diameter: 54.66447979209916
D_MAX - D_DP = -0.45244372347517015 mm
Tomato No.: 5
Delta P: 2.2066535607665023 , std: 0.0 ; Inferred diameter: 49.79050061691561
Time of contact: 5.5270491000010225 , std: 0.0 ; Inferred diameter: 50.52709179103316
D_TC - D_DP = 0.7365911741175495 mm
Delta P MAX: 5.347248005141523 , std: 0.0 Inferred diameter: 49.328728148964764
D_MAX - D_DP = -0.46177246795084415 mm
Tomato No.: 1
Delta P: 2.4741861316109546 , std: 0.0 ; Inferred diameter: 53.09570546560839
Time of contact: 5.0533668999996735 , std: 0.0 ; Inferred diameter: 57.521309963787594
D_TC - D_DP = 4.425604498179204 mm
Delta P MAX: 6.009718904549526 , std: 0.0 Inferred diameter: 54.15041263025708
D_MAX - D_DP = 1.0547071646486899 mm
Tomato No.: 2
Delta P: 2.192907085037834 , std: 0.0 ; Inferred diameter: 49.62364303444661
Time of contact: 5.5901849000001675 , std: 0.0 ; Inferred diameter: 49.596551153775096
D_TC - D_DP = -0.027091880671513024 mm
Delta P MAX: 5.277999429126098 , std: 0.0 Inferred diameter: 48.8428200240565
D_MAX - D_DP = -0.7808230103901082 mm
Tomato No.: 3
Delta P: 3.138431097911929 , std: 0.0 ; Inferred diameter: 61.108532575148644
Time of contact: 4.7368066999998851 , std: 0.0 ; Inferred diameter: 62.37304283736031
D_TC - D_DP = 1.2645102622116653 mm
Delta P MAX: 6.951218302653118 , std: 0.0 Inferred diameter: 61.07590798811316
D_MAX - D_DP = -0.03262458703548532 mm
Tomato No.: 4
Delta P: 2.5692156575922502 , std: 0.0 ; Inferred diameter: 54.27115522594848
Time of contact: 5.178124400001252 , std: 0.0 ; Inferred diameter: 55.56974985759884
D_TC - D_DP = 1.29859463165036 mm
Delta P MAX: 6.135077650344801 , std: 0.0 Inferred diameter: 55.068402734898875
D_MAX - D_DP = 0.7972475089503916 mm
Tomato No.: 5
Delta P: 2.689260852396279 , std: 0.0 ; Inferred diameter: 55.72803079169493
Time of contact: 5.092273800000839 , std: 0.0 ; Inferred diameter: 56.91269602156537
D_TC - D_DP = 1.1846652298704399 mm
Delta P MAX: 6.250078033307254 , std: 0.0 Inferred diameter: 55.91116866986649
D_MAX - D_DP = 0.18313787817156424 mm
Tomato No.: 1
Delta P: 3.203597486686919 , std: 0.0 ; Inferred diameter: 61.906589989417775

Time of contact: 4.719190099998741 , std: 0.0 ; Inferred diameter: 62.63745566755094
 D_TC - D_DP = 0.7308656781331635 mm
 Delta P MAX: 7.003471522262689 , std: 0.0 Inferred diameter: 61.4685167443302
 D_MAX - D_DP = -0.4380732450875726 mm
 Tomato No.: 2
 Delta P: 2.6699936790714203 , std: 0.0 ; Inferred diameter: 55.498535738862515
 Time of contact: 5.070821799999976 , std: 0.0 ; Inferred diameter: 57.24826596645138
 D_TC - D_DP = 1.7497302275888629 mm
 Delta P MAX: 6.220159231684274 , std: 0.0 Inferred diameter: 55.691912471118584
 D_MAX - D_DP = 0.19337673225606977 mm
 Tomato No.: 3
 Delta P: 2.2050420502618895 , std: 0.0 ; Inferred diameter: 49.770939767121774
 Time of contact: 5.538547900001504 , std: 0.0 ; Inferred diameter: 50.362507138691946
 D_TC - D_DP = 0.5915673715701715 mm
 Delta P MAX: 5.395366557196841 , std: 0.0 Inferred diameter: 49.66636968252196
 D_MAX - D_DP = -0.1045700845998141 mm
 Tomato No.: 4
 Delta P: 2.6238082740698543 , std: 0.0 ; Inferred diameter: 54.94642827705861
 Time of contact: 5.149743899999521 , std: 0.0 ; Inferred diameter: 56.01370113584959
 D_TC - D_DP = 1.0672728587909788 mm
 Delta P MAX: 6.054946688481856 , std: 0.0 Inferred diameter: 54.481591460882676
 D_MAX - D_DP = -0.4648368161759322 mm
 Tomato No.: 5
 Delta P: 2.489833400229546 , std: 0.0 ; Inferred diameter: 53.28925139733618
 Time of contact: 5.052136800000881 , std: 0.0 ; Inferred diameter: 57.54055220641218
 D_TC - D_DP = 4.251300809076 mm
 Delta P MAX: 6.061621838409696 , std: 0.0 Inferred diameter: 54.530470007544615
 D_MAX - D_DP = 1.2412186102084348 mm
 Tomato No.: 1
 Delta P: 2.196037094521638 , std: 0.0 ; Inferred diameter: 49.661635740786856
 Time of contact: 5.5462325000007695 , std: 0.0 ; Inferred diameter: 50.25251590315367
 D_TC - D_DP = 0.5908801623668154 mm
 Delta P MAX: 5.295832130491544 , std: 0.0 Inferred diameter: 48.96794973763505
 D_MAX - D_DP = -0.6936860031518037 mm
 Tomato No.: 2
 Delta P: 2.677289584629379 , std: 0.0 ; Inferred diameter: 55.58543869049184
 Time of contact: 5.041636300000391 , std: 0.0 ; Inferred diameter: 57.70480972115663
 D_TC - D_DP = 2.1193710306647873 mm
 Delta P MAX: 6.219287249682743 , std: 0.0 Inferred diameter: 55.685522259991764
 D_MAX - D_DP = 0.10008356949992248 mm
 Tomato No.: 3
 Delta P: 2.43061202592405 , std: 0.0 ; Inferred diameter: 52.55672380114352
 Time of contact: 5.023649700000533 , std: 0.0 ; Inferred diameter: 57.98617101009921
 D_TC - D_DP = 5.429447208955693 mm
 Delta P MAX: 5.914976200087306 , std: 0.0 Inferred diameter: 53.45666256456626
 D_MAX - D_DP = 0.8999387634227389 mm
 Tomato No.: 4
 Delta P: 3.1827490761781987 , std: 0.0 ; Inferred diameter: 61.65127081499397
 Time of contact: 4.744849500000782 , std: 0.0 ; Inferred diameter: 62.25232604120473
 D_TC - D_DP = 0.6010552262107609 mm
 Delta P MAX: 7.021307913588661 , std: 0.0 Inferred diameter: 61.60253190102702
 D_MAX - D_DP = -0.04873891396695029 mm
 Tomato No.: 5
 Delta P: 2.6001196451819713 , std: 0.0 ; Inferred diameter: 54.65341625935657
 Time of contact: 5.1528839000002336 , std: 0.0 ; Inferred diameter: 55.964582656120946
 D_TC - D_DP = 1.3111663967643779 mm
 Delta P MAX: 6.049165385988353 , std: 0.0 Inferred diameter: 54.43925808009718
 D_MAX - D_DP = -0.2141581792593854 mm

REVIEWERS' COMMENTS

Reviewer #1 (Remarks to the Author):

The revised version has been greatly improved by adding additional modeling and computing the expected resolution for different fluid actuator systems with varying ranges of pressures.

I did review references [23] and [27], but still disagree with your assessment. Reference [23] uses both pressure and strain independently, showing that one can be inferred from the other. The signal for the pressure line alone is clearly shown and demonstrates how interactions can be inferred. As the sensor is inside the pressure lines on an external PCB, it also does not require a redesign - the strain sensor has not been used here. The sensor used in [23] is the US9111, which is available from Alibaba for \$2. Reference [27] also treats the pressure sensors as an independent information source. They are using the MPXV6115V for \$20 inside the pressure line (not inside the actuator), and combine this information with resistor sensing readings on a host PC.

Studying this problem with more accurate (and an order of magnitude more expensive sensors) and thoroughly classifying is an important addition to the literature, but still makes this paper incremental in my opinion.

Reviewer #2 (Remarks to the Author):

My comments and suggestions are adequately addressed to improve the contribution, significance and versatility of this study. I do not have any other suggestions.

Reviewer #3 (Remarks to the Author):

The authors addressed all my comments with additional experiments and explanations.

I recommend this paper for publication as it is.

Response to the Reviewers' Comments for Manuscript: “A Retrofit Sensing Strategy for Soft Fluidic Robots”

In the following, we address each of the Reviewers' comments/suggestions (*in italic*). The revisions are highlighted in blue.

Response to Reviewer #1

Comment 1A

The revised version has been greatly improved by adding additional modeling and computing the expected resolution for different fluid actuator systems with varying ranges of pressures.

We thank the reviewer for recognizing the improvement of our manuscript with additional modeling and experimental work during the last revision.

Comment 1B

I did review references [23] and [27], but still disagree with your assessment. Reference [23] uses both pressure and strain independently, showing that one can be inferred from the other. The signal for the pressure line alone is clearly shown and demonstrates how interactions can be inferred. As the sensor is inside the pressure lines on an external PCB, it also does not require a redesign - the strain sensor has not been used here. The sensor used in [23] is the US9111, which is available from Alibaba for \$2. Reference [27] also treats the pressure sensors as an independent information source. They are using the MPXV6115V for \$20 inside the pressure line (not inside the actuator), and combine this information with resistor sensing readings on a host PC. Studying this problem with more accurate (and an order of magnitude more expensive sensors) and thoroughly classifying is an important addition to the literature, but still makes this paper incremental in my opinion.
newline

We must respectfully disagree with the Reviewer. Our aim, as explicitly stated in the original manuscript, is to gain a better understanding of the underlying principles that allow for fluidic self-sensing and make it applicable for a wide range of soft fluidic actuators. As also mentioned in our article, fluidic self-sensing has been previously explored in the literature (as also indicated by Reviewer #2 who acknowledged our clear description of the literature). In the revised manuscript, we reveal that the change in pressure-volume response of a soft fluidic actuator during activation can be used to infer interactions with the environment, and in particular that this sensing principle can be applied to a wide range of soft fluidic actuators without the need for design changes. The basic yet powerful principles shown in this work make it possible to bring (some) sensing capabilities

to most soft fluidic devices without the need for design changes, which we think makes a significant contribution to the field of self-sensing in fluidic actuators (as also recognised by Reviewer #2 and #3).

If we compare our aim to Reference [27], we see that in Reference [27] a fluidic tube sensor to measure the interactions between wearable devices and the environment is proposed. Even though the pressure sensor can be used as an independent information source, the proposed sensing device is only a sensor, not an actuator. It therefore does not harness pressure variations of an pneumatic actuator that interacts with the environment to implement self-sensing during actuation. Following the sensing method in [27], one could attach the fluidic tube sensor to another soft fluidic actuator, such as the work in reference [28]. However, sensor/actuator redesign is still necessary when applying the sensing method in [27] to other soft fluidic actuators and the integration will inevitably affect the performance of the actuator.

Moreover, upon a thorough reexamination of reference [23], we also come to the conclusion that in this work the pressure variations alone in combination with the used setup are not used (and enough) to implement self-sensing. We do realize that reference [23] proposes a phase plot method based on both strain and pressure sensors to distinguish the grasping of cylindrical objects with varying diameter (Fig. 15 in [23]) and to infer contact with the environment (Fig. 16 in [23]). It does not present any method that uses the pressure measurement alone to infer the grasp object/external contact. Even though the two pressure-time signals in Fig. 14 in [23] shows a difference between the empty grasp and grasping of 7.4cm object, the paper did not give any explanation about this pressure difference, or mention any method that correlates the sensing target with this pressure signal alone. Actually, this pressure signal difference in Fig. 14 is rather difficult to understand based on the available information in [23]. Since [23] does not mention the type of the compressed air source, we do not know if the air supply is based on pressure control, flow control or something else (which is thoroughly discussed in our work). We guess that [23] uses a pressure controller to inflate the actuator, because [23] mentions in Section V. Experimental Validation that “The actuator was not required to lift the object, but rather to inflate to 41 kPa and hold firmly to the shape of the object”. Then we expect that the pressure controller regulates the pressure inside the actuator always to a same value in the end, regardless of the grasp object. However, the two pressure signals in Fig. 14 in [23] do not reach the same value before the fill valve closes at $t = 3$ s. It still remains unclear how one can obtain useful sensing information with the pressure measurement alone based on the information in [23].

To further indicate the aim of reference [23], reference [23] makes the claim throughout the article that the sensing capabilities are achieved through the combination of both strain and pressure sensors:

“We present a fiber reinforced soft pneumatic actuator with integrated strain and pressure sensors. We demonstrate that combining these sensors into the same actuator enables proprioception of both actuator curvature and environmental contact forces. ... The grasp radius is estimated from the relationship between the sensor pair.” (Abstract)

“We address these limitations in this paper by incorporating a pair of sensors into the actuator (Figure 1) which permit a proprioception mechanism for tighter control of bending and estimation of the success of a grasp.” (I. Introduction)

“The combined sensor readings of strain and pressure are used to determine grasp radius and even to estimate the success of the grasp (Figure 15)” (V. Experimental Validation, A. Grasping an Object)

“We have presented a soft robotic constant curvature actuator with integrated pressure

and strain sensing. Together, these sensors enable a range of sensing capabilities which are new to soft robotics.” (VII. Conclusion)

Even though reference [23] measures pressure and strain independently, it does not show any evidence that one can be inferred from the other when the interaction between the actuator and the environment is unknown. The Fig. 6 in [23] shows the correlation between pressure and curvature, which stands only when the actuator is not interacting with the environment. To sense the grasp object/external contact, [23] proposes the phase plots in Fig. 15 and 16, which constitute both the pressure and strain sensor measurements. It should be noted that in these phase plots, one can not infer the strain based on the pressure measurement, or the other way around, when the grasp object/external contact is unknown. The phase plot method presented in [23] requires both the pressure and strain sensor measurements to sense the grasp object and external contact. Therefore, it is also not meaningful to make a comparison between the cost of the sensor used in the work of reference [23] and our work, even though we are positive that cheaper sensors might be available through different suppliers that could achieve the required sensitivity needed in our self-sensing approach.

Therefore, we believe that our work is not incremental, but gives us a much better understanding of the underlying principles that allow for fluidic self-sensing, and makes it possible to apply for a wide range of (existing) soft fluidic actuators.

Response to Reviewer #2

Comment 2A

My comments and suggestions are adequately addressed to improve the contribution, significance and versatility of this study. I do not have any other suggestions.

We thank the reviewer for recognizing the improvement of our manuscript during the last revision.

Response to Reviewer #3

Comment 3A

The authors addressed all my comments with additional experiments and explanations. I recommend this paper for publication as it is.

We thank the reviewer for recognizing the improvement of our manuscript during the last revision.